

# A paleoprecipitation and paleotemperature reconstruction of the Last Interglacial in the southeastern Alps

Charlotte Honiat[1], Gabriella Koltai[1], Yuri Dublyansky[1], R. Lawrence Edwards[2], Haiwei Zhang[3], Hai Cheng[3], Christoph Spötl[1]

[1]Institute of Geology, University of Innsbruck, 52 Innrain, 6020 Innsbruck, Austria
[2]Department of Earth Sciences, University of Minnesota, Minneapolis, MN 55455, USA
[3]Institute of Global Environmental Change, Xi'an Jiaotong University, Xi'an 710054, China

*Correspondence to*: Charlotte Honiat (charlotte.honiat@student.uibk.ac.at)

## Abstract

The Last Interglacial (LIG, ~130–116 ka) was one of the warmest interglacials of the past 800,000 years and an important test bed for future climate conditions warmer than today. LIG temperature reconstructions from marine records as well as paleoclimate models show that mid and high northern latitudes were considerably warmer by about 2 to 5°C compared to today. In Central Europe, the LIG has been widely studied using pollen and more recently chironomids preserved in lake sediments. While these bio-archives document temperatures changes across the LIG, they are commonly poorly constrained chronologically. Speleothems, and fluid inclusions contained therein, offer superior age control and provide information on past climate, including qualitative and partly also quantitative records of temperature and precipitation. Here, we present a precisely dated fluid inclusion record based on seven speleothems from two caves in the SE Alps (Obir and Katerloch) and use a δD/T transfer function to reconstruct regional LIG temperatures. We report a temperature change across the glacial/interglacial transition of 5.2 ± 3.1 °C, and peak temperatures at ~127 ka of 2.4 ± 2.8 °C above today's mean (1973-2002). The fluid inclusion δD record of these speleothems exhibits millennial-scale events during the LIG that are not well expressed in the $\delta^{18}O_{calcite}$. The early LIG in the SE Alps was marked by an important climate instability followed by progressively more stable conditions. Our record suggests that the SE Alps predominantly received Atlantic-derived moisture during the early and mid LIG, while more Mediterranean moisture reached the study site at the end of the LIG, buffering the speleothem $\delta^{18}O_{calcite}$ signal. The return towards colder conditions is marked by an increase in $\delta^{13}C$ starting at ~118 ka indicating a decline of the vegetation and soil activity.





## 1. Introduction

The Last Interglacial (LIG, also known as Marine Isotope Stage (MIS) 5e or Eemian; ~130 to 115 ka) was the most recent time period before the Holocene when the global climate was as warm or even warmer than today (Fischer et al., 2018;

Otto-Bliesner et al., 2021). Given that modern global temperatures are approaching the warmth of the LIG (Bova et al., 2021), this most recent interglacial prior to anthropogenic impact currently receives substantial interest by the paleoclimate community as it provides an important benchmark for even warmer conditions and a case to study the response of the hydrological cycle to differently distributed radiative forcing (Scussolini et al., 2019). Surface temperature anomalies during the LIG were not evenly distributed around the globe, and some regions, notably the high northern latitudes, experienced a

disproportionally large warming (CAPE-Last Interglacial Project Members, 2006; Thomas et al., 2020).

The most widely available LIG temperature reconstructions include sea-surface temperature (SST) estimates derived from various inorganic and organic proxies of deep-sea sediments (e.g., Martrat et al., 2007; Tzedakis et al., 2018) with substantial differences due to incoherent chronologies (Capron et al., 2017). Climate model simulations of the LIG based on SSTs and ice-core based temperatures show that land masses were considerably warmer by about 2° to 5°C at mid- and high

northern latitudes (Bakker et al., 2014). For Central Europe, previous studies found that summer temperatures may have been about 1–2 °C higher than present-day (Kaspar, 2005; Lunt et al., 2013). Pollen (Kühl and Litt, 2007) and chironomids (Bolland et al., 2021) retrieved from European lake sediments provide constraints on summer (July) air temperatures, but unfortunately lack sufficiently precise age control to define details of the LIG temperature evolution (Govin et al., 2015).

Speleothems are terrestrial archives that can be dated with high accuracy and precision, and different analytical

methods allow to obtain proxy information of past climate (Fairchild & Baker, 2012). Previous speleothem-based studies from Europe provided mostly qualitative temperature information (e.g., Meyer et al., 2008; Moseley et al., 2015; Häuselmann et al., 2015; Vansteenberge et al., 2016). Two recent speleothem studies from Alpine caves used the stable isotopic composition of fluid inclusions to quantitatively constrain the intra-LIG temperature evolution. Both studies showed consistently that the Alps experienced temperatures of up to ~4°C warmer than today (1961–1990) at elevations close to ~2000 m a.s.l. (Johnston et al.,

2018; Wilcox et al., 2020).

In this study we extend our research in the Alps to lower-elevation regions on the southeastern fringe of this mountain range by analysing fluid inclusions – small pockets of drip water trapped in speleothems during their growth (Schwarcz et al., 1976). Paleo-temperature information can be derived from such inclusions by studying: (1)  their stable isotopic composition (e.g., Wainer et al., 2011; Affolter et al., 2019), (2) their homogenisation temperature (e.g., Krüger et al., 2011; Meckler et al.,

2015), and (3) the concentration of noble gases dissolved in the inclusion water (e.g., Kluge et al., 2008; Vogel et al., 2013; Ghadiri et al., 2018). In this study, we use the first, stable isotope-based approach to quantitatively assess the temperature evolution across the LIG based on a set of seven well-dated stalagmites from two caves on the SE fringe of the Alps, Obir and Katerloch. Such physically based paleotemperature data are of particular importance because speleothem proxy data are tightly anchored to a radiometrically determined chronology allowing detailed comparisons across different archives and models.





## 2. Study sites

### 2.1 Obir Caves


The Obir massif (46°30'N,14°23'E) is part of the Northern Karawanken Mountains close to the Austrian-Slovenian border, in the Austrian province of Carinthia (Fig. 1). The eponymous caves open at ~1100 m a.s.l. in Middle Triassic limestone and consist of a series of galleries, chamber and shafts encountered in the 19th century during mining for Pb-

Zn ores that are now connected by artificial galleries. These caves lack natural entrances and were not known prior to the mining activities (which ceased in the early 20[th] century). The caves are of hypogene origin and were formed by aggressive, upwelling $CO_2$-rich groundwater prior to or during the uplift of the Northern Karawanken Mountains (Spötl et al., 2021). Therefore, the Obir caves most likely had only limited air exchange with the outside atmosphere prior to the mining activities. Many parts of these caves are decorated by flowstones, stalactites and stalagmites. For simplicity, the

Obir caves are divided into three main systems: Rasslsystem, Banane system (Fig. A1) and the show cave system.

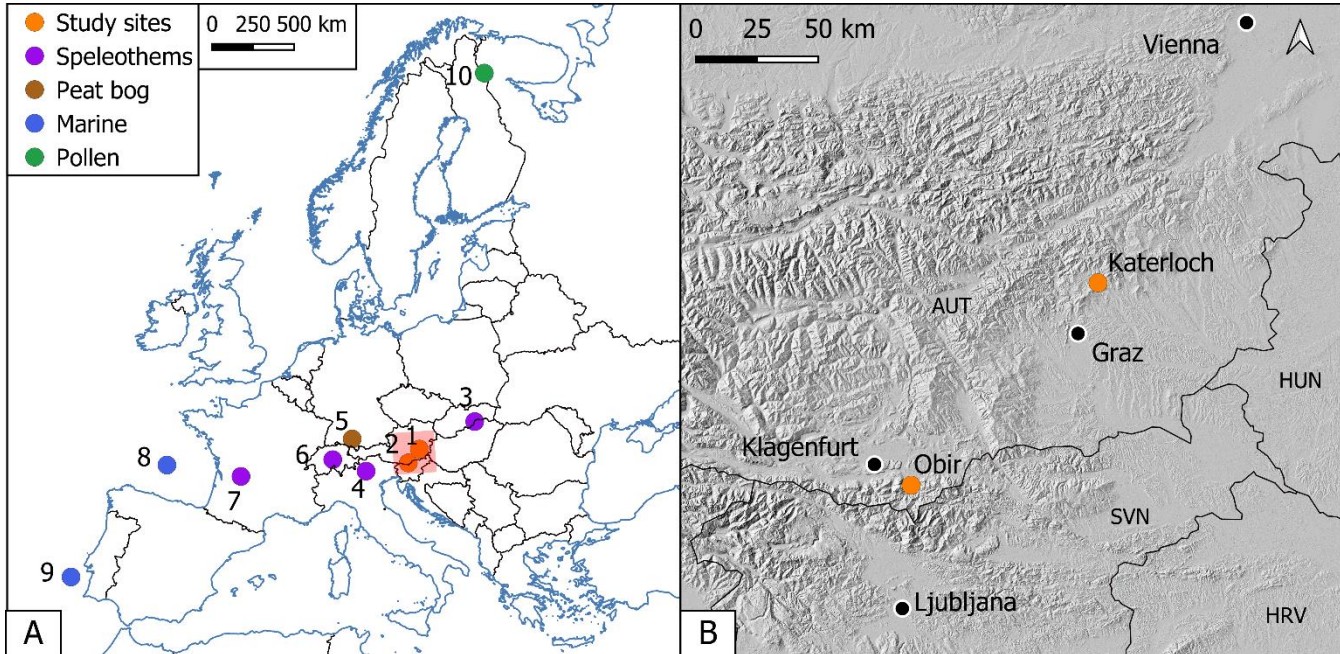

**Figure 1: (A) Location of the study sites (1) Katerloch and (2) Obir and other sites from which paleoclimate data for the LIG are**
**mentioned in the text: (3) Baradla cave (Demény et al., 2017, 2021), (4) Cesare Battisti cave (Johnston et al., 2018), (5) Füramoos (Bolland et al., 2021), (6) Melchsee-Frutt caves (Wilcox et al., 2020), (7) Villars cave (Wainer et al., 2011), (8) deep-sea cores MD04-2845 (Sánchez Goñi et al., 2018; Salonen et al., 2021) and (9) MD01-2444 (Tzedakis et al., 2018), (10) pollen sequence from Sokli (Salonen et al., 2018). The transparent red square in A marks the enlarged digital elevation map shown in B.**





## 2.2 Katerloch cave

The second study site, Katerloch cave, is located in the Austrian province of Styria (47°15'N, 15°32'E), 20 km NNE of the city of Graz and about 115 km from Obir (Fig. 1B). The cave opens in Devonian limestone at an altitude of 901 m a.s.l., follows the general dip of the host rock and comprises a series of halls and narrow restrictions in between. The entrance hall is connected via two short artificial tunnels with a speleothem-rich chamber below, called Phantasiehalle. These two tunnels were blasted during show cave development in the 1950s which probably led to an intensification of the cave ventilation.

## 2.3. Climate at the study sites

Both cave sites receive Atlantic moisture from the W and NW and are also under the influence of Mediterranean air masses from the south, the latter being most pronounced during spring and autumn (including local summer thunderstorms). During the winter season, the North Atlantic Oscillation influences the regional climate on a multi-annual time scale (Boch et al., 2009). The nearest GNIP (Global Network of Isotopes in Precipitation - https://nucleus.iaea.org/wiser) stations are Graz (20 km from Katerloch) and Klagenfurt (15 km from Obir). They provide long (1973 to 2002) time series of stable isotopes in precipitation and air temperature to obtain monthly, seasonal, and long-term $\delta^{18}O/\Delta T$ and $\delta D/\Delta T$ relationships. The two stations receive similar amounts of annual precipitation (810 mm at Graz and 887 mm at Klagenfurt) and their elevations are also comparable (366 and 442 m a.s.l., respectively).

## 2.4 Cave microclimate

The microclimate of both caves has been monitored for many years. In Obir cave the air temperature in the interior parts is stable throughout the year (5.8 ± 0.1°C) and is cooler by ~1°C than the mean annual air temperature (MAAT) of 6.8 ± 0.1°C recorded at the closest weather station of Seeberg (1040 m a.s.l.; ca. 12 km from the cave) (Spötl et al., 2005; Fairchild et al., 2010). Cave air carbon dioxide concentration and its stable C isotopic composition follow a seasonal pattern reflecting today's efficient air exchange with the outside atmosphere through the artificial adits and gives rise to preferred calcite precipitation during winter (Spötl et al., 2005). It is very likely, however, that the air exchange was more restricted prior to the discovery of the caves.

At Katerloch, the air temperature is 4.0°C in Phantasiehalle and 5.7°C in the deepest Seeparadies chamber (Boch et al., 2011). Both temperatures are lower than the MAAT measured near the cave entrance (8.8°C, 2006-2008) and at the weather station of St. Radegund at 725 m a.s.l. ca. 9 km from the cave (8.5 °C; Boch, 2008). This indicates a "cold-trap" behavior of the cave consistent with its sag-type geometry. The cave air circulation was likely weaker in the past, prior to the opening of artificial connections between the large chambers. We therefore consider the temperature of the lowermost chamber (5.7°C) as an approximation of the cave temperature before show cave development.

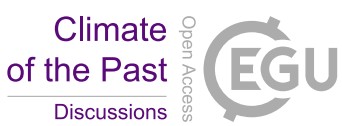

## 2.5 Isotopic composition of drip water

The drip water isotopic composition reflects the meteoric precipitation above the cave but is also affected by processes occurring on the surface (vegetation), in the soil and the epikarst. The long-term (1973-2002) weighted mean of regional meteoric precipitation is -69.8 ± 5.9 for δD and -9.94 ± 0.81 for $δ^{18}O$ at the Klagenfurt GNIP station; and -61.6 ± 6.2 for δD and -8.8 ± 0.7 for $δ^{18}O$ at the Graz GNIP station. In this region the seasonal variability of $δ^{18}O$ and δD has an amplitude of about 5 ‰ and 30 ‰, respectively.

In Obir, the $δ^{18}O$ and δD values of the drip water are fairly constant with mean values of –10.2 ± 0.2 and –68.7 ± 1.6 ‰ VSMOW, respectively (1σ uncertainty; Spötl et al., 2005, and unpublished data by the authors) and lack a seasonal isotopic signal attesting to significant storage and mixing in the (epi)karst. Drip water $δ^{18}O$ and δD values in Katerloch cave are also relatively constant over the year, showing mean values of –8.7 ± 0.1 and –57.5 ± 1.4 ‰ VSMOW, respectively (Boch, 2008).

## 3 Methods

### 3.1 Sampling

        In the Obir caves, two stalagmites (OBI98, 99) from Perlenhalle were retrieved with a hammer and a chisel (Rasslsystem; Fig. A1) following reconnaissance drilling (Spötl and Mattey, 2012) and $^{230}Th$ dating. One broken stalagmite from the Indische Grotte (part of the show cave, OBI14) was at our disposal already, and the other two stalagmites were found broken in the Banana system (OBI117, 118). In Katerloch, stalagmites K2 and K4 were found broken in Phantasiehalle. Their

top parts were missing. See Honiat et al. (2022) for more details on these two samples.

### 3.2 Petrography

        Central slabs were cut from all stalagmites, polished and scanned (Figs. A3 and A4). Small blocks for thin sections were cut along the growth axes of stalagmites K4, K2, OBI98, OBI99 and OBI117. Thin sections were examined petrographically using a Nikon Eclipse polarizing microscope. Additional doubly-polished sections about 200 µm thin were

prepared for fluid-inclusion petrography of K2 and OBI99 stalagmites.

### 3.3 $^{230}Th$ dating

        Multiple subsamples were drilled from stalagmites OBI14 (8), OBI98 (10), OBI99 (10), OBI117 (13), OBI118 (7), K2 (9) and K4 (9) (Table 1; see Honiat et al. (2022) for $^{230}Th$ ages for the two Katerloch speleothems). 80 to 150 mg-subsamples were drilled from stalagmite slabs along discrete laminae. U and Th were separated from the carbonate matrix and purified in

a clean-room laboratory. The samples were prepared following the chemistry procedure as described in Edwards et al. (1987). The measurements were performed using multicollector inductively coupled plasma mass spectrometer (ThermoFisher Neptune Plus, Bremen, Germany) at the University of Minnesota, USA, and at the Xi'an Jiatong University, China, using the





technique described by Cheng et al. (2013). Depth-age models were constructed using OxCal (version 4.4) and a Poisson-process deposition model (Bronk Ramsey, 2008; Bronk Ramsey and Lee, 2013).

**3.4 Calcite stable isotopes**

Subsamples for stable isotope analyses were taken along the growth axes of all stalagmites either using a hand-held drilling device or a Merchantek micromill. OBI98 and OBI99 were drilled at 2 mm increments, while the sampling resolution of OBI118 was 1 mm. Stalagmites OBI117 and OBI14 were micromilled at 0.2 mm resolution. The isotope analyses were performed using a Delta V Plus isotope ratio mass spectrometer linked to a Gasbench II (both ThermoFisher, Bremen, Germany) following the procedure reported by Spötl (2011). Calibration of the instrument was done by using international reference materials and the results are reported in per mil relative to Vienna Pee Dee Belemnite (VPDB). Long-term precision at the 1-sigma level is 0.06‰ and 0.08‰ for $\delta^{13}C$ and $\delta^{18}O_{calcite}$, respectively. The two stalagmites from Katerloch were sampled and analysed in the same laboratory as reported by Honiat et al. (2022).

**3.5 Fluid inclusion stable isotopes**

The stable isotopic composition of stalagmite fluid inclusion water was analysed using a Delta V Advantage isotope ratio mass spectrometer following crushing and high-temperature conversion as described by Dublyansky and Spötl (2009). A total of 28 subsamples (0.4 to 3.0 g) were cut from OBI99, 22 from OBI117, 16 from OBI118, 4 from OBI14, 3 from OBI98, 17 from K2 and 20 from K4. δD values are reported in per mil relative to Vienna Standard Mean Ocean Water (VSMOW). The average long-term precision of replicate measurements of our in-house calcite standard (a low-temperature calcite spar) is 1.5 ‰ for δD for water amounts between 0.2 and 1 μL. In order to be compared to modern-day precipitation the δD values were corrected for the global ice-volume effect of 0.064‰ per meter of sea-level rise (Duplessy et al., 2007) using global sea-level data (Rohling et al., 2019).

**4. Results**

**4.1 Petrography**

The Obir and Katerloch stalagmites consist of coarsely crystalline elongated columnar calcite. In Katerloch, distinct macroscopic lamination is noticeable, consisting of white, porous laminae rich in aqueous inclusions formed during summer alternating with translucent and more compact laminae formed during winter (Boch et al., 2011). No petrographic evidence of hiati was observed in the Katerloch samples. The fabric of the Obir stalagmites is compact columnar and lamination is hardly visible. Petrographic hiati are locally present; in OBI117 these are marked by thin micrite layers and the presence of opaque



organic inclusions (Fig. A5). A hiatus in OBI14 is marked by a slight change in color (pale yellow calcite), and by a less translucent layer rich in detritus in OBI118.




**Figure 2: Transmitted-light images of fluid-inclusion assemblages in the studied speleothems. a) Primary fluid inclusions along growth layers in stalagmite OBI99, b) intracrystalline fluid inclusions in OBI99, c) primary inclusion-rich and inclusion-poor growth layers in stalagmite K2, d) inter-crystalline elongated inclusions in sample K2. The white arrow indicates the growth direction.**

Primary single-phase fluid inclusions were observed in both K2 and OBI99 stalagmite samples. Fluid inclusions in K2 are inter-crystalline and elongated (cf. Kendall and Broughton, 1978) in the compact laminae (~100 to 150 μm in length; Fig. 2d) and large interconnected elongate (up to 500 μm; Fig. 2c) in the white porous laminae. OBI99 contains fewer fluid inclusions that are concentrated along growth layers (Fig. 2a). These inclusions are smaller (10 to 30 μm; Fig. 2b), intracrystalline, and rounded or pyriform in shape (rounded part oriented towards the base of the layer and a spike pointing in





the growth direction - cf. Lopez-Elorza et al., 2021). Petrographic observations showed that the fluid inclusions in our samples
are primary in origin, well preserved, and suitable for bulk instrumental analyses of FI water stable isotope composition.

## 4.2 Geochronology

All samples show low U concentrations (between 70 and 280 ppb) but also little detrital Th content allowing precise
dating with relative age uncertainties of 0.4–1.6% (Table 1). Depth-age models are provided in Figure 3. The average growth
rate of the Katerloch stalagmites (0.4 mm a$^{-1}$ - Honiat et al., 2022) is significantly higher than that of the Obir stalagmites
(from 0.003 to 0.1 mm a$^{-1}$).

**Figure 3: Depth-age models for Obir (OBI) and Katerloch (K) stalagmites. The average growth rates are indicated.**


The OBI14 LIG record started at 130.2 ± 0.5 ka and the first growth episode lasted until 119.9 ± 0.5 ka. After a short
hiatus, growth continued from 119.4 ± 0.7 ka to 112.6 ± 0.7 ka but with a slower growth rate (Fig. 3). Stalagmite OBI98 started





at 126.4 ± 1.0 ka with a short segment of slow growth (0.01 mm a$^{-1}$) followed by a longer interval of faster growth (0.1 mm a$^{-1}$) and terminated with a short final segment of again slower growth at 119.5 ± 1 ka (Fig. 3). OBI99 started growing at 129.9

± 0.6 ka and stopped at 120.1 ± 1.7 ka, with a slow-growing section between 129.4 ± 0.8 ka and 126.2 ± 0.6 ka. OBI117 started growing at 171.7 ± 2.8 ka and shows several hiati until 116.5 ± 1.1 ka. The oldest part of the record is characterised by a slow growth rate from 135.0 ± 0.8 ka to 130.3 ± 0.9 ka (Fig. 3). Growth accelerated after a first hiatus (127.9 ± 0.9 ka to 129.2 ± 0.8 ka) and remained constant after a second hiatus (124.2 ± 0.7 ka to 124.6 ± 0.5 ka), and finally slowed down after a third hiatus (116.5 ± 1.1 ka to 121.1 ± 0.7 ka).

The record of OBI118 (whose base is missing) started at 126.4 ± 0.9 ka and lasted until 121.8 ± 0.4 ka. Following a short hiatus, the growth rate diminished (121.2 ± 1.1 ka to 119.5 ± 0.5 ka) and continued to slow down until the end of the record at 115.0 ± 1.3 ka (Fig. 3).





| Sample Number | $^{238}U$ (ppb) | $^{232}Th$ (ppt) | $^{230}Th / ^{232}Th$ (atomic x10$^{-6}$) | $\delta^{234}U^*$ (measured) | $^{230}Th / ^{238}U$ (activity) | $^{230}Th$ Age (yr) (uncorrected) | $^{230}Th$ Age (yr) (corrected) | $\delta^{234}U_{Initial}^{**}$ (corrected) | $^{230}Th$ Age (yr BP)$^{***}$ (corrected) |
|---|---|---|---|---|---|---|---|---|---|
| OBI14-155 | 98.6 ±0.2 | 278 ±6 | 4380 ±94 | 140.8 ±2.1 | 0.7494 ±0.0022 | 113016 ±696 | 112947 ±697 | 194 ±3 | 112878 ±697 |
| OBI14-177 | 89.7 ±0.1 | 224 ±5 | 4894 ±101 | 117.3 ±1.8 | 0.7413 ±0.0018 | 115505 ±613 | 115442 ±615 | 162 ±3 | 115373 ±615 |
| OBI 14-184 | 82.8 ±0.1 | 238 ±5 | 4356 ±90 | 128.0 ±1.3 | 0.7582 ±0.0019 | 117898 ±597 | 117827 ±598 | 178 ±2 | 117756 ±598 |
| OBI-19-192 | 100.3 ±0.2 | 650 ±13 | 1910 ±41 | 102.4 ±3.3 | 0.7500 ±0.0060 | 121108 ±1896 | 120943 ±1897 | 144 ±5 | 120872 ±1897 |
| OBI14-194 | 89.8 ±0.1 | 253 ±5 | 4413 ±95 | 106.7 ±1.7 | 0.7557 ±0.0019 | 121777 ±674 | 121705 ±676 | 150 ±2 | 121636 ±676 |
| OBI14-225 | 117.6 ±0.1 | 258 ±5 | 5696 ±117 | 109.7 ±1.7 | 0.7593 ±0.0016 | 122149 ±606 | 122093 ±607 | 155 ±2 | 122024 ±607 |
| OBI 14-265 | 135.4 ±0.2 | 132 ±3 | 12976 ±290 | 103.9 ±1.6 | 0.7651 ±0.0020 | 125216 ±708 | 125191 ±708 | 148 ±2 | 125120 ±708 |
| OBI14-295 | 153.5 ±0.2 | 102 ±2 | 19117 ±434 | 106.3 ±1.6 | 0.7725 ±0.0018 | 126848 ±652 | 126831 ±652 | 152 ±2 | 126762 ±652 |
| OBI14-335 | 141.7 ±0.1 | 141 ±3 | 12935 ±275 | 110.3 ±1.5 | 0.7830 ±0.0017 | 129083 ±648 | 129057 ±648 | 159 ±2 | 128988 ±648 |
| OBI14-347 | 123.1 ±0.1 | 195 ±4 | 8420 ±183 | 145.6 ±1.6 | 0.8084 ±0.0018 | 128154 ±647 | 128116 ±647 | 209 ±2 | 128047 ±647 |
| OBI98-11 | 89.8 ±0.1 | 18 ±2 | 61587 ±6341 | 103.2 ±1.6 | 0.7453 ±0.0019 | 119579 ±642 | 119574 ±642 | 145 ±2 | 119505 ±642 |
| OBI98-40 | 124.3 ±0.2 | 98 ±2 | 15336 ±347 | 80.9 ±1.8 | 0.7354 ±0.0017 | 121720 ±655 | 121699 ±655 | 114 ±3 | 121630 ±655 |
| OBI98-90 | 119.9 ±0.1 | 54 ±1 | 27079 ±743 | 77.4 ±1.6 | 0.7368 ±0.0017 | 122989 ±651 | 122977 ±651 | 109 ±2 | 122908 ±651 |
| OBI98-103 | 128.7 ±0.1 | 6 ±2 | 277032 ±88719 | 88.7 ±1.4 | 0.7461 ±0.0015 | 123123 ±552 | 123122 ±552 | 126 ±2 | 123053 ±552 |
| OBI98-161 | 131.8 ±0.2 | 9 ±1 | 180677 ±20301 | 76.0 ±1.6 | 0.7398 ±0.0017 | 124214 ±666 | 124212 ±666 | 108 ±2 | 124143 ±666 |
| OBI98-182 | 147.3 ±0.2 | 15 ±2 | 126769 ±15556 | 184.8 ±1.5 | 0.7698 ±0.0015 | 110076 ±448 | 110073 ±448 | 252 ±2 | 110004 ±448 |
| OBI98-244 | 135.7 ±0.1 | 52 ±1 | 31426 ±874 | 68.5 ±1.4 | 0.7359 ±0.0015 | 124842 ±574 | 124832 ±574 | 97 ±2 | 124763 ±574 |
| OBI98-285 | 146.9 ±0.2 | 9 ±2 | 190830 ±35648 | 85.8 ±1.5 | 0.7462 ±0.0015 | 123808 ±580 | 123806 ±580 | 122 ±2 | 123737 ±580 |
| OBI98-328 | 149.4 ±0.2 | 19 ±1 | 95221 ±4764 | 75.4 ±1.5 | 0.7416 ±0.0015 | 124922 ±603 | 124919 ±603 | 107 ±2 | 124850 ±603 |
| OBI98-335 | 169.1 ±0.2 | 59 ±2 | 35890 ±1266 | 94.4 ±1.6 | 0.7628 ±0.0022 | 126814 ±774 | 126805 ±774 | 135 ±2 | 126736 ±774 |
| OBI98-350 | 72.9 ±0.1 | 2130 ±43 | 552 ±11 | 148.4 ±1.6 | 0.9782 ±0.0025 | 191535 ±1487 | 190855 ±1554 | 254 ±3 | 190786 ±1554 |
| OBI99-22 | 125.4 ±0.1 | 70 ±2 | 22044 ±555 | 99.5 ±1.4 | 0.7466 ±0.0014 | 120763 ±511 | 120748 ±511 | 140 ±2 | 120679 ±511 |
| OBI99-76 | 132.2 ±0.1 | 70 ±2 | 23004 ±573 | 87.4 ±1.5 | 0.7440 ±0.0014 | 122776 ±550 | 122762 ±550 | 124 ±2 | 122693 ±550 |
| OBI99-118 | 120.7 ±0.1 | 13 ±2 | 117987 ±16322 | 97.4 ±1.6 | 0.7539 ±0.0017 | 123388 ±633 | 123385 ±633 | 138 ±2 | 123316 ±633 |
| OBI99-194 | 136.0 ±0.2 | 31 ±1 | 54770 ±2035 | 86.2 ±1.7 | 0.7486 ±0.0015 | 124463 ±617 | 124457 ±617 | 122 ±2 | 124388 ±617 |
| OBI99-250 | 140.0 ±0.2 | 18 ±2 | 95824 ±8716 | 92.9 ±1.6 | 0.7575 ±0.0015 | 125555 ±595 | 125551 ±595 | 132 ±2 | 125482 ±595 |
| OBI99-292 | 130.3 ±0.2 | 37 ±1 | 43690 ±1380 | 90.8 ±1.7 | 0.7575 ±0.0017 | 126067 ±664 | 126059 ±664 | 130 ±2 | 125990 ±664 |
| OBI 99-335 | 129.6 ±0.2 | 137 ±3 | 12020 ±253 | 89.1 ±1.3 | 0.7682 ±0.0021 | 129804 ±744 | 129777 ±744 | 129 ±2 | 129706 ±744 |
| OBI99-370-A | 144.1 ±0.2 | 45 ±1 | 40331 ±1337 | 86.2 ±1.6 | 0.7664 ±0.0016 | 129994 ±663 | 129986 ±663 | 124 ±2 | 129917 ±663 |
| OBI99-375 | 71.5 ±0.1 | 23364 ±468 | 46 ±1 | 114.3 ±1.6 | 0.9036 ±0.0033 | 172192 ±1608 | 163847 ±6112 | 181 ±4 | 163778 ±6112 |
| OBI99-380 | 90.6 ±0.1 | 2932 ±59 | 513 ±10 | 95.5 ±1.7 | 1.007 ±0.0022 | 248091 ±2613 | 247298 ±2653 | 192 ±4 | 247229 ±2653 |
| OBI117-3 | 160.7 ±0.2 | 9081 ±182 | 216 ±4 | 102.3 ±1.7 | 0.7397 ±0.0015 | 118177 ±565 | 116728 ±1166 | 142 ±2 | 116658 ±1166 |
| OBI117-22 | 172.8 ±0.2 | 436 ±9 | 4739 ±96 | 79.4 ±1.7 | 0.7255 ±0.0013 | 119143 ±528 | 119077 ±529 | 111 ±2 | 119006 ±529 |
| OBI117-40 | 154.9 ±0.2 | 6 ±1 | 300207 ±51811 | 81.6 ±1.5 | 0.7331 ±0.0015 | 120895 ±571 | 120893 ±571 | 115 ±2 | 120823 ±571 |
| OBI117-47 | 248.2 ±0.3 | 98 ±2 | 30034 ±672 | 48.8 ±1.4 | 0.7203 ±0.0016 | 124824 ±640 | 124813 ±640 | 69 ±2 | 124743 ±640 |
| OBI117-56 | 272.5 ±0.4 | 101 ±2 | 32208 ±701 | 55.2 ±1.5 | 0.7239 ±0.0016 | 124375 ±624 | 124365 ±624 | 78 ±2 | 124295 ±624 |
| OBI117-59 | 194.9 ±0.3 | 5027 ±101 | 469 ±9 | 49.4 ±1.8 | 0.7330 ±0.0020 | 128824 ±805 | 128121 ±941 | 71 ±3 | 128051 ±941 |
| OBI117-72 | 269.6 ±0.4 | 119 ±3 | 27644 ±596 | 54.9 ±1.6 | 0.7374 ±0.0017 | 128819 ±699 | 128807 ±699 | 79 ±2 | 128737 ±699 |
| OBI117 -76 | 196.6 ±0.2 | 1379 ±28 | 1761 ±35 | 57.8 ±1.3 | 0.7489 ±0.0015 | 131840 ±630 | 131651 ±642 | 84 ±2 | 131582 ±642 |
| OBI117 -103 | 204.1 ±0.3 | 761 ±15 | 3387 ±69 | 75.7 ±1.5 | 0.7664 ±0.0025 | 132796 ±925 | 132698 ±927 | 110 ±2 | 132629 ±927 |
| OBI117-116 | 231.1 ±0.5 | 262 ±5 | 11322 ±237 | 89.1 ±2.3 | 0.7790 ±0.0034 | 133298 ±1288 | 133268 ±1288 | 130 ±3 | 133197 ±1288 |
| OBI117-130 | 152.4 ±0.2 | 445 ±9 | 4740 ±96 | 89.4 ±1.8 | 0.8398 ±0.0015 | 155124 ±861 | 155049 ±861 | 138 ±3 | 154978 ±861 |
| OBI117 -138 | 145.3 ±0.2 | 228 ±5 | 9381 ±194 | 110.9 ±1.6 | 0.8917 ±0.0023 | 168385 ±1188 | 168346 ±1188 | 178 ±3 | 168277 ±1188 |
| OBI117-154 | 181.2 ±0.5 | 900 ±19 | 3016 ±63 | 141.8 ±5.4 | 0.9090 ±0.0034 | 163466 ±2423 | 163348 ±2421 | 225 ±9 | 163278 ±2421 |
| OBI118-4 | 146.3 ±0.1 | 1626 ±33 | 1058 ±21 | 74.3 ±1.3 | 0.7134 ±0.0016 | 116737 ±538 | 116443 ±575 | 103 ±2 | 116372 ±575 |
| OBI118-24 | 160.8 ±0.2 | 327 ±7 | 5722 ±117 | 58.3 ±1.3 | 0.7065 ±0.0018 | 118295 ±614 | 118240 ±615 | 81 ±2 | 118169 ±615 |
| OBI118-40 | 173.6 ±0.2 | 2668 ±53 | 763 ±15 | 51.0 ±1.6 | 0.7113 ±0.0014 | 121459 ±586 | 121040 ±654 | 72 ±2 | 120969 ±654 |
| OBI118-76 | 181.8 ±0.2 | 231 ±5 | 9185 ±189 | 48.5 ±1.6 | 0.7078 ±0.0015 | 120985 ±591 | 120950 ±592 | 68 ±2 | 120879 ±592 |
| OBI118-88 | 194.2 ±0.2 | 106 ±2 | 21437 ±480 | 48.1 ±1.3 | 0.7097 ±0.0013 | 121665 ±509 | 121649 ±509 | 68 ±2 | 121578 ±509 |
| OBI118-118 | 211.6 ±0.2 | 123 ±3 | 20142 ±439 | 41.0 ±1.4 | 0.7091 ±0.0015 | 123198 ±588 | 123182 ±588 | 58 ±2 | 123111 ±588 |
| OBI118-134 | 225.1 ±0.2 | 101 ±2 | 25967 ±573 | 40.6 ±1.2 | 0.7095 ±0.0012 | 123463 ±480 | 123451 ±480 | 57 ±2 | 123380 ±480 |
| OBI118-177 | 253.5 ±0.4 | 201 ±4 | 14872 ±308 | 42.0 ±1.8 | 0.7171 ±0.0015 | 125535 ±686 | 125513 ±686 | 60 ±3 | 125442 ±686 |
| OBI118-Btm | 221.1 ±0.6 | 757 ±16 | 3470 ±73 | 48.4 ±3.4 | 0.7207 ±0.0028 | 125063 ±1246 | 124969 ±1247 | 69 ±5 | 124899 ±1247 |

U decay constants: $\lambda_{238}$ = 1.55125x10$^{-10}$ (Jaffey et al., 1971) and $\lambda_{234}$ = 2.82206x10$^{-6}$ (Cheng et al., 2013). Th decay constant: $\lambda_{230}$ = 9.1705x10$^{-6}$ (Cheng et al., 2013).

*$\delta^{234}U$ = ([$^{234}U/^{238}U$]$_{activity}$ − 1)x1000. ** $\delta^{234}U_{initial}$ was calculated based on $^{230}Th$ age (T), i.e., $\delta^{234}U_{initial}$ = $\delta^{234}U_{measured}$ x $e^{\lambda_{234}xT}$.

Corrected $^{230}Th$ ages assume the initial $^{230}Th/^{232}Th$ atomic ratio of 4.4 ± 2.2 x10$^{-6}$. Those are the values for a material at secular equilibrium, with the bulk earth $^{232}Th/^{238}U$ value of 3.8. The errors are arbitrarily assumed to be 50%.

***B.P. stands for "Before Present" where the "Present" is defined as the year 1950 A.D.






**Table 1: $^{230}$Th dating results of Obir stalagmites. Ages are given in years BP with 2σ uncertainties. For dating results of Katerloch stalagmites see Honiat et al. (2022).**

**4.3 Calcite stable isotopes**

**4.3.1 Oxygen isotopes**

The five Obir stalagmites yielded a well replicated $\delta^{18}O_{calcite}$ record for the LIG. Obir and Katerloch stalagmites also agree in their overall pattern, although the latter exhibit a higher-frequency variability (Fig. 4b). Only stalagmite K4 recorded the onset on the LIG, which is marked by a 2.5‰ rise in $\delta^{18}O_{calcite}$. The same jump in isotope values is observed in stalagmite OBI117, although the actual glacial-interglacial transition is not recorded due to the presence of a hiatus (Fig. 4b). During the LIG only small-scale variations (~0.5‰) of $\delta^{18}O_{calcite}$ values are observed (Fig. 4b). The mean $\delta^{18}O_{calcite}$ values for the interval

126 to 120 ka when all Obir stalagmite records overlap are -7.2 ± 0.3‰ for OBI14, -7.9 ± 0.3‰ for OBI98, -7.8 ± 0.2‰ for OBI99, -7.9 ± 0.2‰ for OBI117 and -7.6 ± 0.2‰ for OBI118. In Katerloch, K2 and K4 show the same mean $\delta^{18}O_{calcite}$ value of -7.6 ± 0.5‰ for the interval where they overlap (126.8 to 128.6 ka).







**Figure 4: a) Hydrogen isotopic composition of fluid inclusion water of stalagmites from Obir and Katerloch caves corrected for the ice-volume effect, b) oxygen and c) carbon isotopic composition of the calcite, d) modelled** $^{230}$**Th ages of each stalagmite with their 2 sigma uncertainties.**

### 4.3.2 Carbon isotopes

The transition from the penultimate glacial (MIS 6) to the LIG is partially recorded by $\delta^{13}$C values in stalagmites K4 and OBI117 and is more abrupt than the oxygen isotope shift recorded by stalagmite K4. The latter stalagmite started growing 129.6 ± 0.4 ka ago with $\delta^{13}$C values of about −6‰. Shortly after the rise in $\delta^{18}O_{calcite}$, there is a 4‰ drop in registered by $\delta^{13}$C.





Stalagmite K2 grew between 128.6 ± 0.5 and 125.0 ± 0.7 ka. During this time period, carbon isotope values are stable, in
agreement with those of stalagmite K4, and lack a long-term trend (Fig. 4c). The mean δ¹³C value of K2 and K4 for the interval
where the two records overlap (128.6-126.8 ka) is −9.7 ± 0.7 ‰.

The Obir stalagmites show a gradual decrease in δ¹³C from ~ -7‰ at 135 ka to ~ -10‰ at 125 ka (Fig. 4c). Only
small-scale variations of up to ~0.5‰ are observed during the LIG. The carbon isotope values of the five Obir stalagmites are
in good agreement from 130 ka until 118 ka. Between 118 ka and 117 ka the values start to rise and are well replicated (within
their age model uncertainties) between OBI118, OBI117 and OBI14. At ~115 ka, the δ¹³C values reach and partly exceed pre-
LIG values (Fig. 4c). The mean δ¹³C values of the interval 126 to 120 ka when all Obir stalagmites overlap are -9.9 ± 0.5‰
for OBI14, -9.3 ± 0.3‰ for OBI98, -9.5 ± 0.2‰ for OBI99, -9.4 ± 0.7‰ for OBI117, and -9.3 ± 0.2‰ for OBI118.

### 4.4 Fluid-inclusion isotopes

A total of 115 calcite subsamples were analysed, but a significant proportion of the fluid-inclusion measurements
(n=38 for the Obir dataset) yielded water amounts too small to obtain reliable isotope results (< 0.1 µL; see Fig. A4 for the
location of these samples). On the other hand, two Katerloch samples had to be excluded because of too large analyte volumes
(>1.5 µL; Fig. A4). Almost all Katerloch samples were duplicated or even triplicated. Not every Obir samples could be
duplicated, however, because the replica had low water amounts; and eventually there was insufficient material for sub-
sampling individual layers.

δD values of sub-samples of K4 and K2 with water contents of 0.1 to 1 µL replicated within 1.5‰. Obir samples,
however, are characterised by generally low and variable amounts of water and the replicated samples yielded a mean standard
deviation of ±2.1‰ for δD. We assign this value to individual measurements and also use it as an uncertainty estimate (Table
A2).

In terms of water content, the measured fluid-inclusion data from both caves lack a long-term trend across the LIG.
We also analysed three Holocene stalagmites for comparison (OBI12 for Obir; K1 and K3 for Katerloch; Table A1).
Fluid-inclusion data of modern calcite were already available for Obir cave (sample OBI1; Dublyansky and Spötl, 2009).

The δD values of the cave drip water agree with the amount-weighted δD mean of modern precipitation for Obir
(Table 2) and only a slight difference is seen for Katerloch. The δD fluid-inclusion values for the LIG optimum (128-125 ka)
were comparable to those of modern day at Obir and are more negative for Katerloch (Table 2). Two samples of a Holocene
stalagmite from Obir (3.6 and 5 ka BP) yielded values that are more negative than modern precipitation and LIG optimum
samples, but close to the values from the second half of the LIG (125-115 ka). Two early Holocene Katerloch samples (11.3
and 9.6 ka) also yielded more negative values than modern precipitation (Table 2). Fluid-inclusion data for penultimate glacial
calcite are comparable between the two cave sites and more negative than modern values.





| GNIP station | Klagenfurt (442 m a.s.l.) | Graz (366 m a.s.l.) |
|---|---|---|
| Amount-weighted mean (1973-2002) | -69.8 ± 5.9 | -61.6 ± 6.2 |
| Study site | Obir caves, 1100 m a.s.l. | Katerloch cave, 901 m a.s.l. |
| Cave drip water | -68.7 ± 0.8 (Fairchild et al., 2010) | -57.5 ± 1.4 (Boch, 2008) |
| Cave pool water | -70.1 ± 0.3 (Dublyansky and Spötl, 2009) | -57.4 ± 1.4 (Boch, 2008) |
| FI of late Holocene speleothem | OBI1 pool spar -70.0 ± 0.6 | n.a. |
| FI of early to mid-Holocene speleothems | -80.4 ± 4.1 (3.6 ka)  -84.4 ± 2.2 (5 ka) | -71.6 ± 1.5 (11.3 ka)  -70.4 ± 2.6 (9.6 ka) |
| FI of LIG optimum (128-125 ka) | -71.1 ± 2.1  to -72.3 ± 2.1 | -62.8 ± 1.5  to -72.2 ± 1.5 |
| FI of second half of the LIG (125-115 ka) | -73.1 ± 2.1  to -82.3 ± 3.0 | n.a. |
| FI of penultimate glacial | -85.1 ± 3.1 to -93.2 ± 2.1 | -83.1 ± 1.5 to -89.7 ± 1.5 |

**Table 2: Summary of fluid inclusion (FI) stable isotope data (δD, ‰ VSMOW) of LIG stalagmites compared to precipitation data from the closest GNIP stations, cave drip water and FI data from Holocene speleothems. n. a.: not available.**

## 5. Discussion

### 5.1 Reliability of the calcite stable isotope record

#### 5.1.1 Oxygen isotopes

The oxygen isotopic composition of drip water in caves is controlled by different factors such as the oceanic moisture source(s), trajectories of the air masses, altitude of cloud condensation, evapotranspiration in the catchment and the temperature in the cave (Rozanski et al., 1992; McDermott, 2004; Lachniet, 2009). In Obir and Katerloch caves $\delta^{18}O$ values of the drip water (−10.2 ± 0.2‰ and −8.7 ± 0.1‰, respectively) are closely related to the $\delta^{18}O$ values of local meteoric precipitation (mean $\delta^{18}O$ values of −9.8‰ at the Klagenfurt station and −8.8‰ at the Graz station), which principally originates from the Atlantic

with a Mediterranean imprint (slightly enriched $\delta^{18}O$ values) (Sodemann and Zubler, 2010). The overall oxygen isotope pattern of Obir and Katerloch stalagmites is similar to that of LIG speleothems from other parts of the Alps (Moseley et al., 2015; Wilcox et al., 2020; Luetscher et al., 2021) which also receive predominantly Atlantic-derived moisture, and where $\delta^{18}O_{calcite}$ primarily reflects atmospheric temperature. The average LIG $\delta^{18}O_{calcite}$ values of the Katerloch and Obir speleothems are also comparable to those of speleothems in the Italian Alps (Johnston et al., 2018, 2021) and in north-eastern Hungary (Demény et

al., 2017, 2021), areas that also receive significant moisture from the Western Mediterranean resulting in slightly enriched



$\delta^{18}O$ values of drip water compared to sites on the northern side of the Alps. The mean LIG $\delta^{18}O_{calcite}$ values of Katerloch and Obir speleothems during the LIG are more depleted than the modern ones.

Oxygen isotope samples along single growth laminae (Hendy test) of Obir and Katerloch stalagmites show constant values, supporting calcite precipitation close to O isotopic equilibrium. In addition, the $\delta^{18}O_{calcite}$ signal is well replicated between the five Obir stalagmites for the time interval they overlap, and likewise for the two Katerloch stalagmites.

### 5.1.2 Carbon isotopes

The carbon isotope signal in speleothems is primarily controlled by vegetation, carbon dynamics in the soil, cave ventilation and associated kinetic isotope fractionation, and possible prior calcite precipitation in the vadose zone (Fairchild et al., 2006). Although seepage waters in Katerloch cave originate from a well-mixed karst aquifer and thus do not transmit a seasonal signal, a seasonal cycle is observed in the calcite fabric and the C isotopic composition of the stalagmites (Boch et al. 2011). In this cave, the seasonally changing air flow exerts a strong control on the drip water chemistry and hence lamina development, resulting in a white porous inclusion-rich and low-$\delta^{13}C$ lamina in summer and a more compact, high-$\delta^{13}C$ lamina in winter (Boch et al., 2011). Furthermore, Boch et al. (2009, 2011) performed Hendy tests on calcite from the top of actively growing stalagmites and calcite precipitated on glass plates and observed an enrichment in $^{13}C$ of up to 4‰ with increasing distance from the central axis, suggesting some kinetic isotope fractionation.

In Obir caves, the seasonally changing ventilation also forces degassing of carbon dioxide during the cold season resulting in enhanced carbon isotope fractionation. This is reflected by $^{13}C$ enrichment in winter calcite (Spötl et al., 2005).

In summary, although subject to kinetic fractionation in the cave on an intra-annual scale, soil bioproductivity exerts a strong first-order control on longer-term carbon isotope variations in Katerloch and Obir speleothems. In addition, anthropogenic interference (mining at Obir and show-cave development at Katerloch) have likely intensified air exchange between the outside atmosphere and the cave interior at both sites, leading to enhanced degassing and hence kinetic carbon isotope fractionation compared to the LIG.

### 5.2 Paleothermometry using fluid-inclusion stable isotope data

### 5.2.1 Constraining paleotemperatures using a combination of $\delta^{18}O_{calcite}$ and $\delta D$

In order to obtain paleotemperatures, only the stalagmite $\delta D$ values were used because the $\delta^{18}O_{FI}$ values in speleothems are influenced by nonclimatic parameters (e.g., kinetic isotope fractionation - Affolter et al., 2019). Several studies suggested that $\delta^{18}O_{FI}$ values may also undergo isotope exchange with the host calcite (e.g., Demény et al., 2016). Thus, we consider $\delta D$ to be a more robust proxy of paleotemperature as there are no other sources of hydrogen once the water entrapped in the calcite. We are confident that our $\delta D$ values are reliable for several reasons: (i) a large majority of the measurements is





replicated (up to 4 times), (ii) δD values are also replicated between coeval stalagmites, and (iii) these data are replicated between the two cave sites located ~115 km apart.

The δD values (after correction for sea level and elevation) were converted to $\delta^{18}O_{FI-Calculated}$ using the local meteoric water line (LMWL) from Klagenfurt for Obir (~15 km from Obir) and from Graz for Katerloch (20 km from Katerloch) which

are the nearest stations of the Austrian Network of Isotopes in Precipitation (ANIP; Hager and Foelsche, 2015) with an observation period of 29 years (1973 to 2002). Temperatures were calculated based on equations of Friedman and O'Neil (1977), Kim and O'Neil (1997), Coplen (2007) and Tremaine et al. (2011), using $\delta^{18}O_{calcite}$ and $\delta^{18}O_{FI-Calculated}$. The equations of Friedman and O'Neil (1977) and Kim and O'Neil (1997) gave realistic temperatures for the LIG for both caves, but unrealistically high temperatures for the penultimate glacial suggesting cave air temperatures of up to ~10°C at 134 ka for Obir

and up to 25°C at 129.5 ka for Katerloch (Fig. 5). The equation of Coplen et al. (2007) yielded unrealistically high temperatures for both Obir and Katerloch LIG records. We therefore do not consider paleotemperature assessments based on the water-calcite isotope equilibrium reliable.

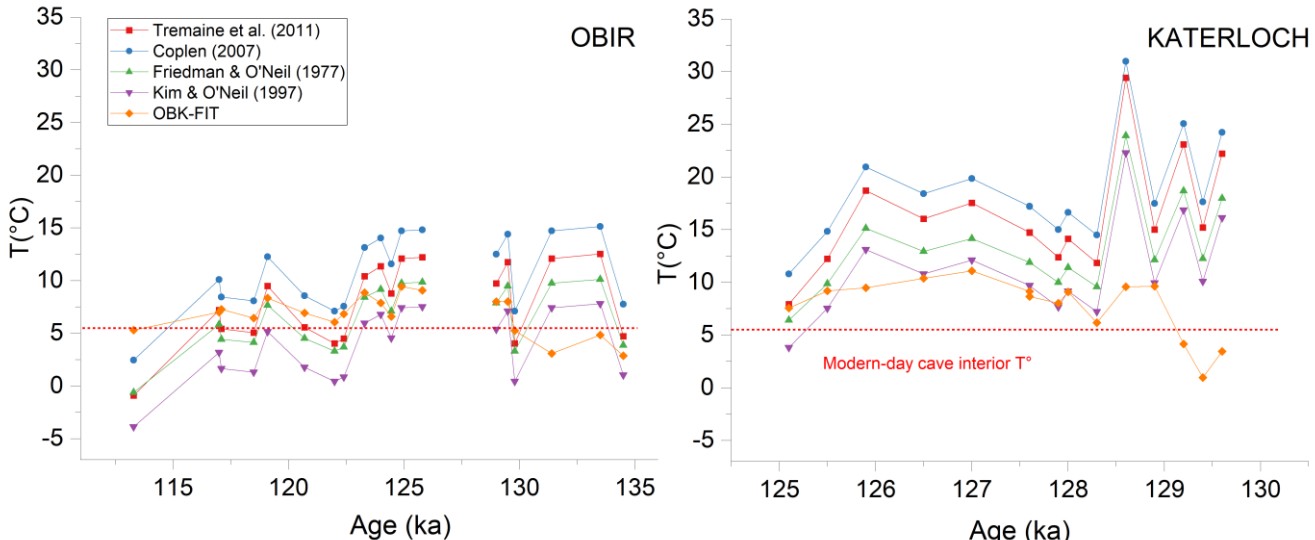

**Figure 5: Results of paleotemperature calculations for Obir and Katerloch using the LMWL of Klagenfurt and Graz, respectively, based on δD data converted to $\delta^{18}O_{FI-Calculated}$ using four different water-calcite isotope fractionation equations (in red, blue, green and purple). The orange data show the results of the water isotope-air temperature relationship δD/T (OBK-FIT). The red dashed line indicates the modern cave interior temperature.**

**5.2.2 Water isotope-air temperature relationship**

We investigated the temperature dependence of the hydrogen (and oxygen) isotope composition of precipitation water in the study region (i.e., multi-annual modern-day δD/T and $\delta^{18}O$/T gradients, respectively). This relationship was investigated by e.g. Rozanski et al. (1992) for Central Europe and applied by Affolter et al. (2019) for a 14 ka-long record from Milandre cave (Switzerland). This approach was also applied to a LIG record from Alpine caves in Switzerland (Wilcox et al., 2020).



The relationship between mean annual δ¹⁸O of precipitation and mean annual air temperature (δ¹⁸O/T) is 0.43 ± 0.18

‰/°C for Klagenfurt (δD/T determined using the LMWL: 3.35 ± 1.40 ‰/°C) and 0.32 ± 0.15 ‰/°C for Graz (δD/T: 2.66 ± 1.25 ‰/°C) (1973–2002; Hager and Foelsche, 2015). Compared to the average European δ¹⁸O/T gradient of 0.59 ± 0.08 ‰/°C (Rozanski et al., 1992) the gradient for Klagenfurt is within combined uncertainties, but that for Graz is significantly smaller. The coefficient of correlation between MAAT and weighted mean δ¹⁸O annual values at Graz and Klagenfurt sites is small (R² <0.2) in comparison to R²=0.54 of Rozanski et al. (1992). We attribute this to the pronounced seasonality in precipitation.

Because it is unclear which δD/T transfer function is appropriate for the LIG, and possible changes in vapor source regions should be considered, we evaluated a range of δD/T relationships (named OBK-FIT), considering both the Klagenfurt and Graz empirical gradients. The OBK-FIT transfer function is anchored at the modern MAAT outside of the cave (6.8 ± 1 °C for Obir; 8.8 ± 1 °C for Katerloch). The modern δD values were corrected for the elevation difference relative to the GNIP stations of Klagenfurt (~650 m from Obir) and Graz (~450 m from Katerloch) assuming a LIG lapse rate identical to the

modern mean for the Austrian Alps of ~0.2‰/100 m for δ¹⁸O, i.e. ~1.6‰/100 m for δD (cf. Poage et al., 2001) and annotated δD_modern. We use the mean weighed δD values from the two nearest GNIP stations instead of the δD drip water values obtained during a few years of cave monitoring, because longer-term monitoring at the GNIP stations provides more robust and coherent relationships. The error of the δD, δD_modern, δD/T, MAAT values outside the cave, and the slope of the LMWL were propagated through the different calculation steps and resulted in a combined paleotemperature uncertainty between 2.1 and 4.5°C. As the

uplift since the LIG in this area is negligible (Sternai et al., 2019) no correction was applied.

**5.3 Millennial-scale variability in LIG European speleothem fluid-inclusion records**

Fluid-inclusion records of LIG speleothems from Europe are scarce (Wainer et al., 2011a; Johnston et al., 2018) and very few proxy records cover the full duration of the LIG (Demény et al., 2017; Wilcox et al., 2020). An interesting first observation is that the δD variability of published records and our record is more pronounced than the variability of the

corresponding δ¹⁸O_calcite records and documents a series of millennial-scale intra-LIG events (Fig. 6).

The Obir-Katerloch record shows a δD rise of up to 25‰ across the glacial-interglacial transition. The same amplitude was reported from caves on Melchsee-Frutt (Swiss Alps) and from Baradla cave, Hungary (Fig. 6). Shortly after the onset of the LIG a drop of about ~10‰ in δD is captured in our record at ~128.3 ± 0.5 ka, in agreement with low δD values in the Hungarian record. A second cooling event is observed at ~124.5 ± 0.5 ka in our record and is coherent with the expansion of

cold water masses in the North Atlantic related to disruptions of the Atlantic Meridional Overturning Circulation (AMOC; Irvalı et al., 2016). The first event in our record was possibly related to cold event C28, a hypothesized Atlantic Ocean meltwater event at ~128.5 ka (Tzedakis et al., 2018). We correlate the second cooling event to C27, which was also identified in speleothems from Melchsee-Frutt (Swiss Alps) between 125.8 ± 0.5 and 124.6 ± 1.0 ka (Wilcox et al., 2020), consistent with our chronology (Fig. 6). The Bigonda speleothem record from the Italian Prealps also suggests a cooling at 124.1 ± 1.8

ka (Fig. 6). This cold event is now well represented in central Europe and is thought to have been an analogue of the 8.2 ka event during the Holocene (Nicholl et al., 2012; Zhou and McManus, 2022). The agreement between our speleothem record



and Atlantic deep-sea sediments record emphasizes that the Atlantic Ocean was the predominant moisture source for our study area also during the LIG. Moreover, during this interglacial speleothem records from the SE Italian Alps (Johnston et al., 2018, 2021) show lower $\delta^{18}O_{calcite}$ values. These authors proposed that a northward shift of the Intertropical Convergence Zone may

have allowed more East Atlantic moisture to cross North Africa before turning northwards into the Mediterranean and Adriatic Seas and reaching the Alps from the south. Our SE Alps records show equally low $\delta^{18}O_{calcite}$ (>~1‰ compared to modern day), thus adding qualitative support to this model.

        The two cold events bracket a climatic optimum from ~127.5 ± 0.5 ka to ~125.5 ± 0.5 ka marked by the highest δD values in all five published European fluid-inclusion records (Fig. 6). Except for Villars cave, this thermal optimum is less

marked in the $\delta^{18}O_{calcite}$ values of these speleothems (Fig. 6). The variability of both δD and $\delta^{18}O_{calcite}$ values in all records decreases after this optimum showing a slowly decreasing trend until 115 ka, suggesting that this was supra-regional signal across large parts of Europe.



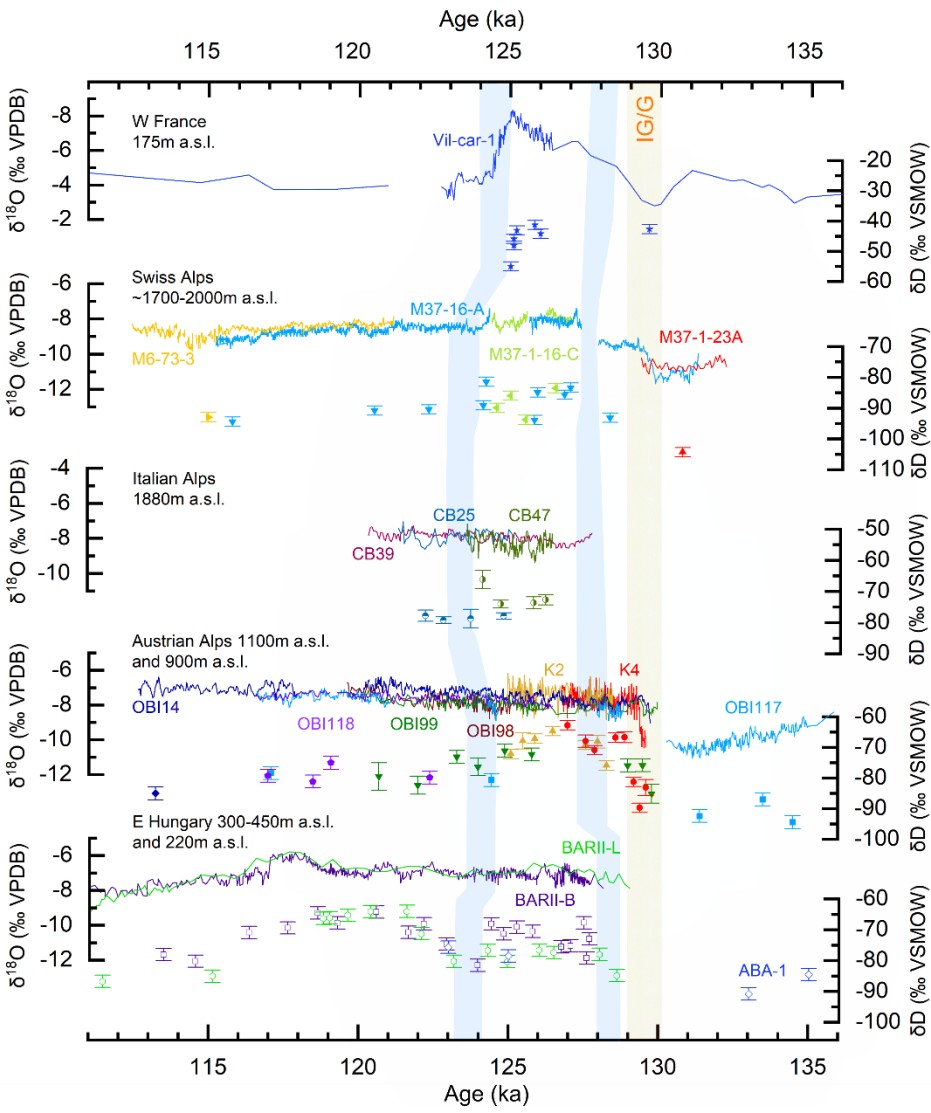

**Figure 6: Comparison of fluid-inclusion speleothem records arranged top down from west to east in Europe. From top to bottom: Villars cave, France (45°30'N, 0°50'E) (Wainer et al., 2011b), Neotektonik cave and Schratten cave on Melchsee-Frutt, Switzerland (46°47'N, 8°16'E) (Wilcox et al., 2020), Cesare Battisti cave, Italy (Johnston et al., 2018) (46°30'N, 11°02'E), Obir (46°30'N, 14°23'E) and Katerloch (47°15'N, 15°32'E) caves (this study), Baradla (48°28'N,20°30'E) and Abaliget (46°8'N, 18°7'E) caves, Hungary (Demény et al., 2017, 2021). The blue colored bars represent cooling events, the yellow bar is the glacial-interglacial transition (IG/G).**

Climate instability during the LIG has also been detected in other European speleothem records which do not include fluid-inclusion isotope data (Drysdale et al., 2009, Regattieri et al., 2014) but their proxy signal is often muted (e.g., Couchoud et al., 2009; Vansteenberge et al., 2019). This is well documented for Alpine speleothems where $\delta^{18}O_{calcite}$ records commonly show only a small variability during the LIG (Moseley et al., 2015; Wilcox et al., 2020; Luetscher et al., 2021; Honiat et al.,





2022). This suggests that the $\delta^{18}O_{calcite}$ signal may be less sensitive to millennial-scale variability during interglacials than the $\delta D$ data of paleo-drip water. More data from other regions (or archives) are needed to explore this further. In this respect it is noteworthy that the $\delta^{18}O_{calcite}$ records of most Alpine speleothems studied so far do not show a strong shift at the end of the LIG. One possible explanation for the $\delta^{18}O_{calcite}$ values to remain at the high interglacial level at the end of the LIG is an increase in the contribution of Mediterranean-sourced moisture at the expense of Atlantic-derived moisture (cf. Johnston et al.,

2021). In contrast, the glacial inception is well recorded by the $\delta^{13}C$ values, starting to increase at ~118 ka in all Alpine speleothem records (Fig. 4 and Wilcox et al., 2020), reflecting a major change in vegetation composition across this mountain range as a result of a lowering of the treeline, and a concomitant decrease in soil bioproductivity.

**5.4 Paleotemperature reconstructions for the LIG in Europe**

The OBK-FIT data indicate a temperature rise at the onset of the LIG of ~ 5.2 ± 3.1° C. After the glacial-interglacial

transition an early warm phase occurred from 129.0 to 128.6 ka, followed by a short and rapid cooling event. This first warm phase is well represented in SST reconstructions from the Iberian margin and in speleothems from Baradla cave (the later record is compromised by a major hiatus) and corresponds to a hiatus in Alpine speleothems from Switzerland (Fig. 7). In the Sokli record from Finland (Salonen et al., 2018), whose chronology is tuned to Alpine (for the onset of the LIG; Moseley et al., 2015) and Belgian speleothems (for the demise of the LIG; Vansteenberge et al., 2016), this initial warming occurred at

130.9 ± 1 ka. A summer temperature reconstruction using chironomids from Füramoos (Bolland et al., 2021) in the northern Alpine foreland shows an unconformity in the early LIG and was tuned to marine records, rendering a detailed comparison difficult.

The OBK-FIT temperatures reached their maximum between 127.5 and 125.5 ka in agreement with the SKR-FIT record from Switzerland (Wilcox et al., 2020) and a temperature reconstruction from deep-sea sediments in the Bay of Biscay

(45°N; Sánchez Goñi et al., 2018; Salonen et al., 2021; Fig. 7). We therefore regard this period as the thermal optimum with temperatures possibly ~2°C higher than modern day (1973-2002) at our sites (2.4 ± 2.8°C for OBK-FIT; 900-1100 m a.s.l.). The SKR-FIT record from the Swiss Alps indicates temperatures up to 4.3 ± 1.4 °C higher than modern-day (1971–1990) between 127.3 ± 0.7 and 125.9 ± 0.5 ka (Wilcox et al., 2020) at ~1800 m a.s.l., and the record from Cesare Battisti (Italian Alps) indicates a +4.3 ± 1.6 °C temperature anomaly at ~2000 m a.s.l. for the period of 126.0–125.3 ka with respect to 1961–

1990. The climate of the LIG in the Alps, but also at Baradla cave (Hungary) and at the core site MD04-2845 in the Atlantic Ocean became cooler after ~124 ka with mean temperatures close to today's values and a lower temperature variability than during the first half of the LIG. In the lacustrine chironomid record from Füramoos located north of the Alps, a decline of the summer temperature from ~15.5°C during mid-LIG to 12 °C during the late LIG was associated with the decreasing Northern Hemisphere July insolation (Bolland et al., 2021). After 118 ka, temperatures slowly fell below the modern-day values at our

study sites, suggesting a gradual rather than an abrupt onset of the glacial inception. This gradual cooling was also captured by the Swiss speleothems and in the SSTs from the Bay of Biscay and the Iberian Margin, while a more pronounced cooling is suggested by the Hungarian speleothem record (Fig. 7).



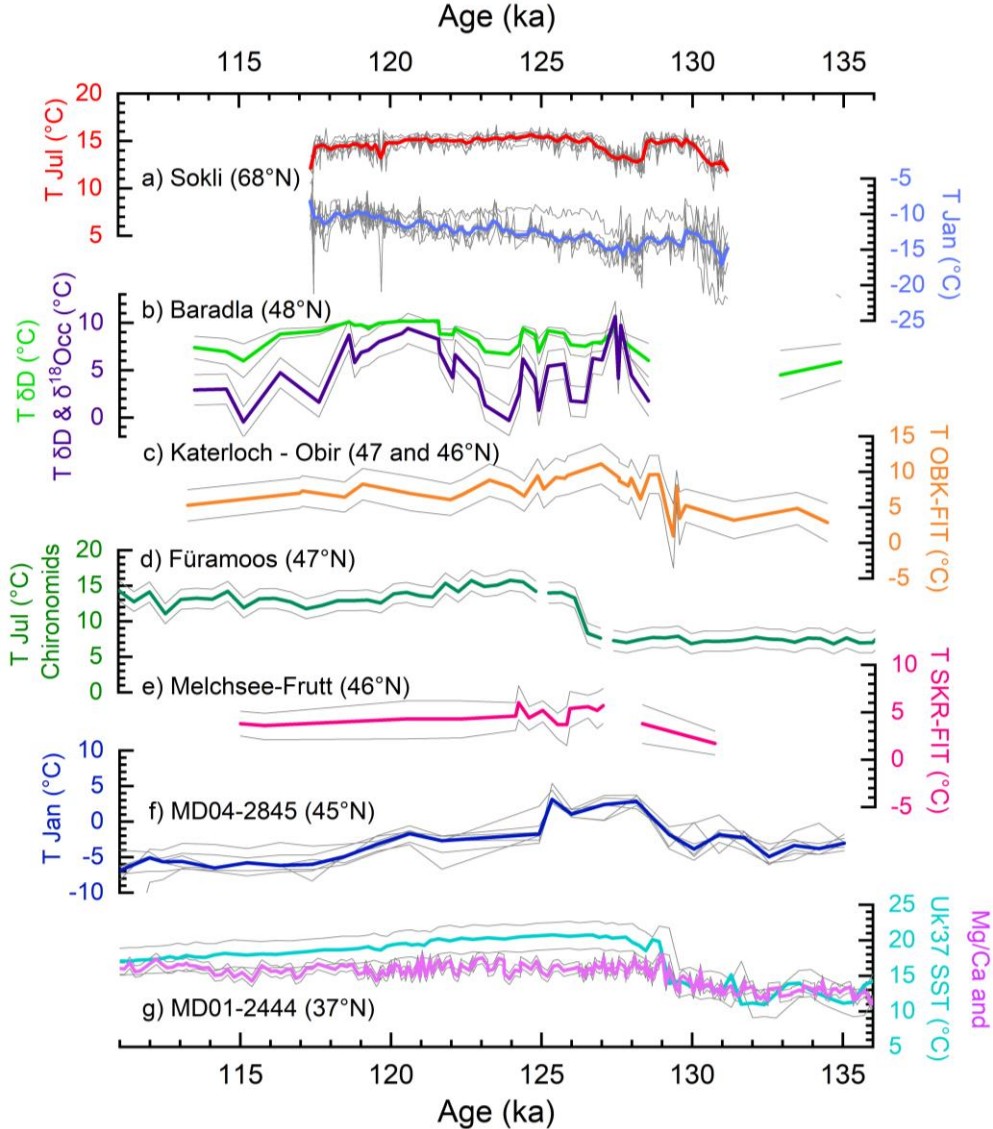


**Figure 7: Comparison of different paleotemperature reconstructions for the LIG in Europe arranged along a N-S transect and plotted on their published time scales: a) pollen-based mean July and January temperature reconstructions from Sokli, Finland (Salonen et al., 2018), b) temperature reconstruction based on speleothem fluid-inclusion data from Baradla cave, Hungary, using the δD transfer function (green) and the calcite-water oxygen isotope thermometer based on δD and $\delta^{18}O_{calcite}$ data (purple; although**

**the authors suggested this reconstruction is not robust), (Demény et al., 2021), c) speleothem fluid-inclusion data using the OBK-FIT data (this study), d) chironomid-based mean July temperature from Füramoos, Germany (Bolland et al., 2021), e) speleothem fluid-inclusion data (using the SKR-FIT data) from Switzerland (Wilcox et al., 2020), f) pollen-based mean January temperature**



**reconstruction from deep-sea core MD04-2845** (Sánchez Goñi et al., 2018; Salonen et al., 2021)**, and g) reconstructions of January sea-surface temperatures (SST) for deep-sea core MD01-2444, derived from Mg/Ca and alkenone data** (Tzedakis et al., 2018)**. The thin**
**grey lines represent the results from different calibration models for the records from Sokli and core MD04-2845; they represent the error envelopes of the temperature estimates.**

## 6. Conclusions

The Obir and Katerloch speleothems provide a well-replicated and precisely dated record of paleotemperatures in the
SE Alps during the LIG. The regional warming at the glacial-interglacial transition determined using a δD/T fluid-inclusion transfer function (OBK-FIT) was $5.2 \pm 3.1$ °C. The early part of the LIG (~129 to 124 ka) was marked by peak-warm conditions interrupted by short cooling events likely related to meltwater discharge events in the North Atlantic. We report temperatures up to $+ 2.4 \pm 2.8$°C higher than modern-day (1973 to 2002) during the LIG optimum at ~127 ka. Temperatures then slightly decreased during the mid-LIG (124 to 121 ka) and gradually dropped below modern-day temperatures after about 118 ka. The
combination of δD and $\delta^{18}O_{calcite}$ proxy data suggests that during the early and mid LIG the SE Alps received predominantly moisture from the Atlantic Ocean while the proportion of Mediterranean-derived moisture increased towards the end of the LIG, buffering the $\delta^{18}O_{calcite}$ signal.





**Appendices**

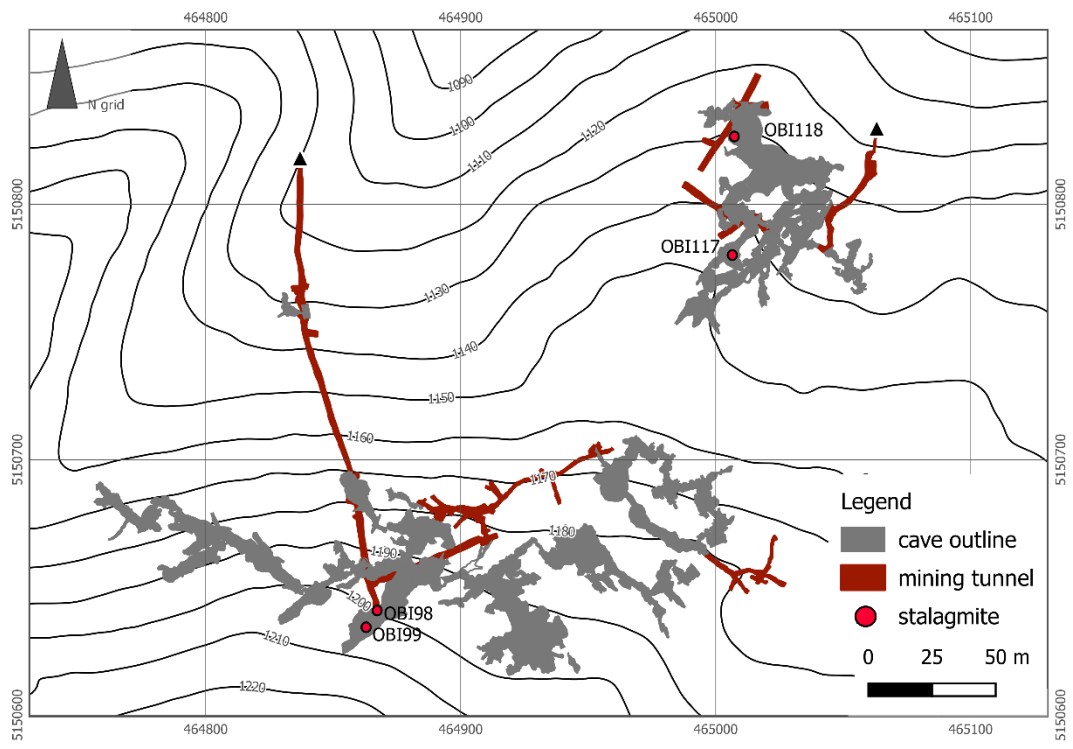


**Fig. A1: Map of the Rasslsystem (left) and the Banane system (right) of the Obir caves showing the locations of the studied stalagmites. Sample OBI14 was found in the Indische Grotte of the show cave part (not shown).**






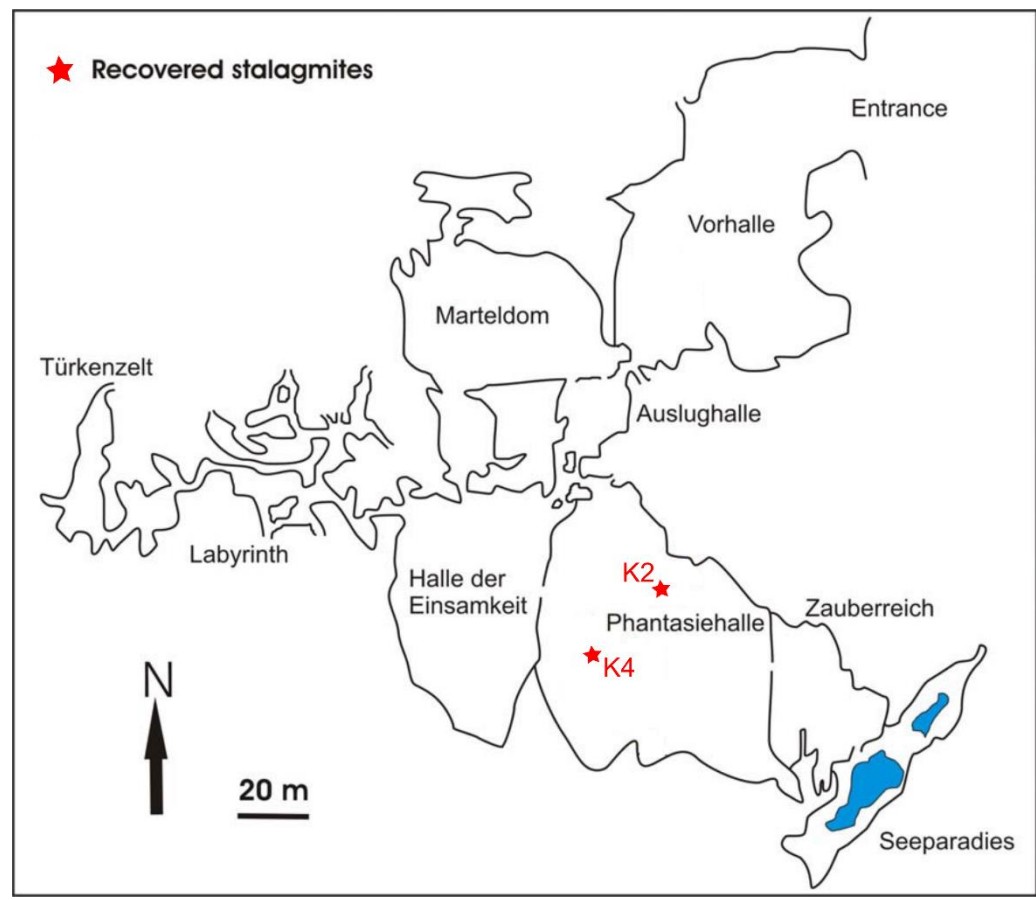

**Fig. A2: Plan view of Katerloch showing the locations of recovered stalagmites (red stars). Map modified after Boch et al. (2011).**






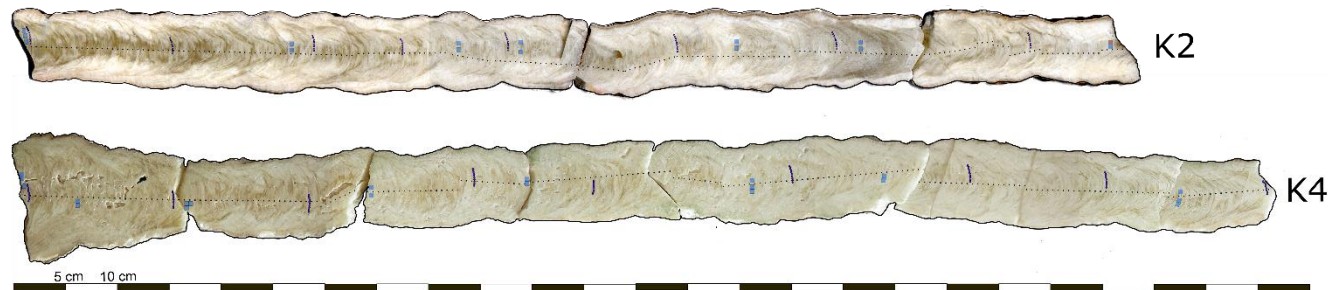

**Fig. A3: Scans of the longitudinal cross-sections of the Obir cave stalagmites showing hand-drilled (thin black dotted lines) and micromilled traces for stable isotopes (vertical grey bars), samples for $^{230}$Th dating (purple dots), and fluid inclusions (blue squares (data presented in Table 1) and orange squares (samples yielding <0.1 μL of water). Note that the fluid-inclusions samples were taken on the opposite half of the respective stalagmite slabs.**

**Fig. A4: Scan of the longitudinal cross-sections of the studied Katerloch stalagmites, with the sampling trace for stable isotopes (dotted black lines), $^{230}$Th dating (purple dots), fluid inclusions (blue squares; data presented in Table 1). Note that the fluid-inclusion samples were taken on the opposite half of the respective slabs.**







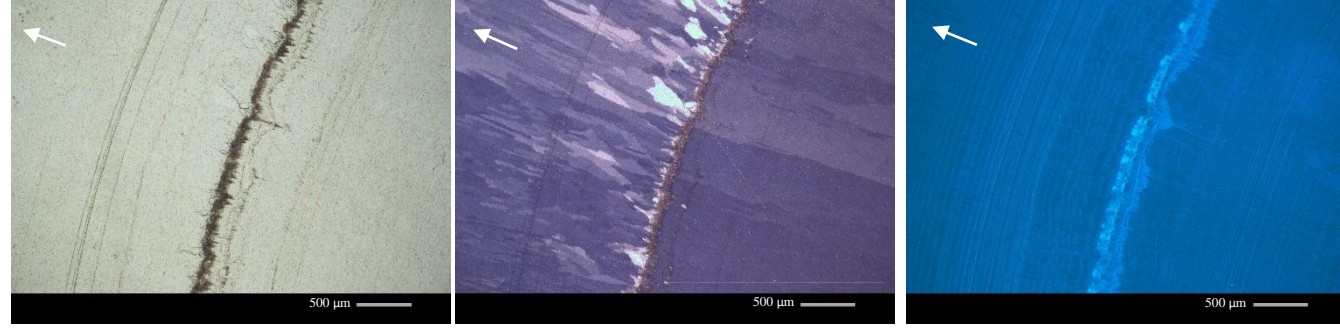


**Fig. A5: Thin-section photomicrographs of the hiatus between 130.3 ± 0.9 ka and 129.2 ± 0.8 ka in stalagmite OBI117. Note nucleation event followed by re-growth of calcite crystals. Parallel (left) and crossed nicols (middle) and epifluorescence image (right). The white arrows indicate the growth direction.**







| Speleothem Sample | Water amount (µL) | Water content (µL/g) | δD (‰VSMOW) measured | Mean δD (‰ VSMOW) | δD (SD) | δD error | Correction for RSL (m) | Mean δD (‰)* adjusted for RSL | Age (ka) | Time span ± (ka) |
|---|---|---|---|---|---|---|---|---|---|---|
| OBI99-362-358-A | 0.26 | 0.19 | -86.9 | -84.7 | 3.1 | 3.1 | -6.5 | -85.1 | 129.8 | 0.04 |
| OBI99-362-358-C | 0.26 | 0.19 | -82.5 | | | | | | 129.8 | 0.04 |
| OBI99-340-345-A | 0.40 | 0.23 | -75.2 | -75.9 | 2.1 | 2.1 | 11.2 | -75.9 | 129.5 | 0.04 |
| OBI99-340-345-B | 0.11 | 0.10 | -78.0 | | | | | | 129.5 | 0.04 |
| OBI99-329-323-A | 0.16 | 0.13 | -76.4 | -76.4 | N/A | 2.1 | 7.0 | -75.9 | 129.0 | 0.23 |
| OBI99-272-267-A | 0.30 | 0.22 | -74.1 | -72.8 | 1.8 | 2.1 | 8.0 | -72.3 | 125.8 | 0.05 |
| OBI99-272-267-B | 0.73 | 0.59 | -70.7 | | | | | | 125.8 | 0.05 |
| OBI99-272-267-C | 0.19 | 0.16 | -73.7 | | | | | | 125.8 | 0.05 |
| OBI99-222-217-A | 1.19 | 1.20 | -71.7 | -71.3 | 0.5 | 2.1 | 3.0 | -71.1 | 124.9 | 0.06 |
| OBI99-222-217-B | 1.21 | 1.19 | -71.4 | | | | | | 124.9 | 0.06 |
| OBI99-222-217-C | 1.40 | 1.40 | -70.8 | | | | | | 124.9 | 0.06 |
| OBi99-162-167-A | 0.12 | 0.12 | -78.2 | -76.1 | 2.9 | 2.9 | -3.9 | -76.4 | 124.0 | 0.05 |
| OBI99-162-167-C | 0.39 | 0.2 | -74.1 | | | | | | 124.0 | 0.05 |
| OBI99-123-118-A | 0.42 | 0.46 | -72.6 | -72.8 | 0.3 | 2.1 | -4.6 | -73.1 | 123.3 | 0.03 |
| OBI99-123-118-B | 0.41 | 0.45 | -73.0 | | | | | | 123.3 | 0.03 |
| OBI99-56-62-A | 0.85 | 0.59 | -83.9 | -81.8 | 3.0 | 3.0 | -8.6 | -82.3 | 122.0 | 0.10 |
| OBI99-56-62-C | 0.17 | 0.25 | -79.7 | | | | | | 122.0 | 0.10 |
| OBI99-23-18-A | 0.21 | 0.22 | -82.1 | -78.9 | 4.5 | 4.5 | -9.0 | -79.5 | 120.7 | 0.06 |
| OBI99-23-18-B | 0.23 | 0.24 | -75.7 | | | | | | 120.7 | 0.06 |
| OBI117-3-8-C | 0.23 | 0.46 | -75.8 | -77.2 | 1.9 | 2.1 | -18.4 | -78.4 | 117.1 | 0.34 |
| OBI117-3-8-B | 0.19 | 0.35 | -78.6 | | | | | | 117.1 | 0.34 |
| OBI117-45-55 | 0.78 | 0.26 | -80.4 | -80.4 | N/A | 2.1 | -4.6 | -80.7 | 124.5 | 0.14 |
| OBI117-80-87-A | 0.11 | 0.04 | -90.9 | -91.1 | N/A | 2.1 | -20.0 | -92.1 | 131.4 | 0.33 |
| OBI117-101-108-B | 0.29 | 0.15 | -81.7 | -82.6 | 1.2 | 2.1 | -62.0 | -86.5 | 133.5 | 0.43 |
| OBI117-101-108-A | 0.78 | 0.41 | -83.9 | | | | | | 133.5 | 0.43 |
| OBI117-100-108-C | 1.04 | 0.54 | -81.5 | | | | | | 133.5 | 0.43 |
| OBI117-100-108-D | 0.47 | 0.47 | -83.1 | | | | | | 133.5 | 0.43 |
| OBI117-109-116-A | 0.17 | 0.12 | -88.4 | -88.4 | N/A | 2.1 | -75.0 | -93.2 | 134.5 | 0.47 |
| OBI118-5-10-A | 0.28 | 0.66 | -78.6 | -78.2 | 0.7 | 2.1 | -17.3 | -79.3 | 115.9 | 0.07 |
| OBI118-5-10-B | 0.38 | 0.79 | -77.7 | | | | | | 115.9 | 0.07 |
| OBI118-24-29-D | 0.12 | 0.05 | -80.2 | -80.2 | N/A | 2.1 | -14.8 | -81.2 | 117.9 | 0.07 |
| OBI118-32-37-B | 0.28 | 0.1 | -74.7 | -74.5 | 0.2 | 2.1 | -7.6 | -75.0 | 118.8 | 0.07 |
| OBI118-32-37-D | 0.36 | 0.13 | -74.4 | | | | | | 118.8 | 0.07 |
| OBI118-100-105-A | 0.16 | 0.06 | -79.2 | -79.2 | N/A | 2.1 | -10.8 | -79.9 | 122.4 | 0.08 |
| OBI14-157-162-C | 0.21 | 0.1 | -83.1 | -83.1 | N/A | 2.1 | -30.0 | -85.0 | 113.3 | 0.34 |
| K2-620-610 | 0.63 | 2.43 | -72.4 | -72.4 | N/A | 1.5 | 3.5 | -72.2 | 125.1 | 0.01 |
| K2-880-885-A | 0.2 | 0.48 | -65.4 | -68.1 | 2.7 | 2.7 | 3.5 | -67.9 | 125.5 | 0.01 |
| K2-880-885-B | 0.27 | 0.52 | -70.8 | | | | | | 125.5 | 0.01 |
| K2-1060-1070 -C | 0.2 | 0.49 | -69.7 | -67.6 | 1.4 | 1.5 | 8.0 | -67.1 | 125.9 | 0.04 |
| K2-1060-1070 -A | 0.45 | 0.77 | -65.5 | | | | | | 125.9 | 0.04 |
| K2-1205-1210-A | 0.37 | 1,00 | -64.9 | -65.1 | 0.2 | 1.5 | 6.7 | -64.7 | 126.5 | 0.06 |
| K2-1205-1210-B | 0.28 | 0.91 | -65.3 | | | | | | 126.5 | 0.06 |
| K2-1310-1320-C | 0.47 | 1.08 | -69.3 | -69.5 | 0.2 | 1.5 | 4.0 | -69.3 | 127.6 | 0.02 |
| K2-1310-1320-A | 0.32 | 0.9 | -69.7 | | | | | | 127.6 | 0.02 |
| K2-1440-1445-B | 0.35 | 0.93 | -65.8 | -67.9 | 2.1 | 2.1 | -3.0 | -68.1 | 128.0 | 0.02 |
| K2-1445-1450 | 0.44 | 0.88 | -70.0 | | | | | | 128.0 | 0.02 |
| K2-1587-1600-A | 0.47 | 2.24 | -78.0 | -76.5 | 1.2 | 1.5 | -3.0 | -76.7 | 128.3 | 0.07 |
| K2-1587-1600-B | 0.53 | 2.65 | -75.0 | | | | | | 128.3 | 0.07 |
| K2-1610-1615 | 0.82 | 1.78 | -76.4 | | | | | | 128.3 | 0.07 |
| K4-108-112-C | 0.31 | 0.75 | -62.1 | -63.6 | 1.5 | 1.5 | 12.0 | -62.8 | 127.0 | 0.01 |
| K4-108-112-B | 0.31 | 0.65 | -65.0 | | | | | | 127.0 | 0.01 |
| K4-410-420 | 0.29 | 0.67 | -70.0 | -68.2 | 2.0 | 2.0 | 4.0 | -68.0 | 127.6 | 0.02 |
| K4-415-420 | 0.65 | 1.58 | -66.4 | | | | | | 127.6 | 0.02 |
| K4-545-550-B | 0.28 | 0.59 | -71.0 | -70.9 | 0.1 | 1.5 | -0.6 | -70.9 | 127.9 | 0.01 |
| K4-545-550-C | 0.23 | 0.69 | -70.9 | | | | | | 127.9 | 0.01 |
| K4-780-785-A | 0.73 | 1.5 | -66.6 | -66.8 | 0.2 | 1.5 | 0.0 | -66.8 | 128.6 | 0.02 |
| K4-780-785-B | 0.46 | 1.25 | -67.1 | | | | | | 128.6 | 0.02 |
| K4-918-922-A | 0.34 | 0.63 | -68.9 | -67.1 | 1.8 | 1.8 | 6.0 | -66.7 | 128.9 | 0.01 |
| K4-918-922-B | 0.45 | 0.91 | -65.2 | | | | | | 128.9 | 0.01 |
| K4-1080-1085-A | 0.32 | 1.20 | -80.7 | -81.7 | 0.9 | 1.5 | 7.0 | -81.3 | 129.2 | 0.02 |
| K4-1080-1085-B | 0.22 | 0.78 | -82.9 | | | | | | 129.2 | 0.02 |
| K4-1085-1090 | 1.37 | 1.95 | -81.6 | | | | | | 129.2 | 0.02 |
| K4-1200-1203-A | 0.12 | 0.59 | -90.1 | -90.4 | 0.4 | 1.5 | 11.2 | -89.7 | 129.4 | 0.01 |
| K4-1200-1203-B | 0.19 | 1.07 | -90.8 | | | | | | 129.4 | 0.01 |
| K4-1250-1247 | 0.53 | 1.91 | -82.2 | | | | | | 129.6 | 0.03 |
| K4-1250-1255 | 0.95 | 2.11 | -80.8 | -83.3 | 2.6 | 2.6 | 3.0 | -83.1 | 129.6 | 0.03 |
| K4-1250-1260-A | 0.34 | 1.30 | -86.9 | | | | | | 129.6 | 0.03 |
| K3-B | 0.67 | 1.37 | -70.5 | -70.5 | 0.1 | 1.5 | -39.4 | -73.0 | 11.3 | N/A |
| K3-A | 0.57 | 0.93 | -70.5 | | | | | | 11.3 | N/A |
| K1-A | 0.67 | 0.91 | -68.2 | -70.0 | 2.6 | 2.6 | -27.3 | -71.7 | 9.6 | N/A |
| K1-B | 0.22 | 0.59 | -71.7 | | | | | | 9.6 | N/A |
| OBI12-100-105-A | 0.70 | 0.33 | -77.2 | -80.1 | 4.1 | 4.1 | -4.1 | -80.4 | 3.6 | 0.10 |
| OBI12-100-105-B | 0.24 | 0.15 | -83.0 | | | | | | 3.6 | 0.10 |
| OBI12-165-170-C | 0.35 | 0.15 | -82.2 | -83.8 | 2.2 | 2.2 | -9.7 | -84.4 | 4.9 | 0.10 |
| OBI12-165-170-B | 0.15 | 0.06 | -85.4 | | | | | | 5.0 | 0.10 |






**Table A1: Results of fluid-inclusion measurements for the LIG stalagmites OBI99, OBI117, OBI118, OBI14, K2 and K4 and the Holocene specimens OBI12, K3 and K1. OBI98 is not included because of too low water content (< 0.1 µL). δD values were corrected for relative sea-level (RSL; 0.064‰ per meter; Duplessy et al., 2007). Time span represents the duration covered by the respective calcite blocks cut from the stalagmites (based on the growth rate) used for the fluid-inclusion measurements but does not take into account the age model uncertainty.**




| Sample ID (In stratigraphic order) | Age (ka) | δD corrected for RSL (‰ VSMOW) | δD corrected error ± (‰) | $T_{LIG}$ (°C) OBK-FIT * | Error ± (°C) |
|---|---|---|---|---|---|
| OBI117-109/116-A | 134.5 | -93.2 | 2.1 | 2.9 | 2.7 |
| OBI117-101/108-B | 133.5 | -86.5 | 2.1 | 4.8 | 2.3 |
| OBI117-80/87-A | 131.4 | -92.1 | 2.1 | 3.1 | 2.6 |
| OBI99-362-358-A | 129.8 | -85.1 | 3.1 | 5.2 | 2.3 |
| K4-1250/1260-A | 129.6 | -83.1 | 2.6 | 3.2 | 3.7 |
| OBI99-340-345-A | 129.5 | -75.9 | 2.1 | 8.0 | 2.2 |
| K4-1200-1203-B | 129.4 | -89.7 | 1.5 | 1.0 | 4.5 |
| K4-1080/1085-B | 129.2 | -81.3 | 1.5 | 4.2 | 3.4 |
| OBI99-329-323-A | 129 | -75.9 | 2.1 | 8.0 | 2.2 |
| K4-918-922-B | 128.9 | -66.7 | 1.8 | 9.7 | 2.7 |
| K4-780/785-B | 128.6 | -66.8 | 1.5 | 9.7 | 2.6 |
| K2-1587/1600-B | 128.3 | -75.9 | 1.5 | 6.2 | 2.9 |
| K2-1445-1450 | 128 | -68.1 | 2.1 | 9.2 | 2.7 |
| K4-545/550-C | 127.9 | -70.9 | 1.5 | 8.1 | 2.6 |
| K2-1310/1320-A | 127.6 | -69.3 | 1.5 | 8.7 | 2.6 |
| K4-415/420 | 127.6 | -68.0 | 2.0 | 9.2 | 2.7 |
| K4-108-112-C | 127 | -62.8 | 1.5 | 11.2 | 2.8 |
| K2-1205/1210-B | 126.5 | -64.7 | 1.5 | 10.5 | 2.7 |
| K2-1060/1070 -A | 125.9 | -67.1 | 1.5 | 9.5 | 2.7 |
| OBI99-272-267-A | 125.8 | -72.3 | 2.1 | 9.1 | 2.3 |
| K2-880/885-B | 125.5 | -67.9 | 2.7 | 9.2 | 2.7 |
| K2-620/625-A | 125.1 | -72.2 | 1.5 | 7.6 | 2.7 |
| OBI99-222-217-A | 124.9 | -71.1 | 2.1 | 9.4 | 2.4 |
| OBI117-45-55 | 124.45 | -80.7 | 2.1 | 6.6 | 2.1 |
| OBI99-162-167-B | 124 | -76.4 | 2.1 | 7.9 | 2.2 |
| OBI99-123-118-A | 123.3 | -73.1 | 2.1 | 8.8 | 2.3 |
| OBI118-100-105-A | 122.4 | -79.9 | 2.1 | 6.8 | 2.1 |
| OBI99-56-62-A | 122 | -82.5 | 3.0 | 6.1 | 2.2 |
| OBI99-23-18-A | 120.7 | -79.5 | 4.5 | 6.9 | 2.4 |
| OBI118-32-37-A | 118.8 | -74.8 | 2.1 | 8.3 | 2.2 |
| OBI118-24-29-D | 117.9 | -81.2 | 2.1 | 6.4 | 2.1 |
| OBI117-3/8-C | 117.1 | -78.4 | 2.1 | 7.3 | 2.1 |
| OBI118-5-10-A | 115.85 | -79.3 | 2.1 | 7.0 | 2.1 |
| OBI14-157/162-C | 113.3 | -85.0 | 2.1 | 5.3 | 2.2 |

$$ * \quad T_{LIG} = T_{modern} - \frac{\delta D_{modern} - \delta D_{FI\ corrected}}{\delta D/T_{gradient}} $$

$T_{modern}$ Obir = 6.8 ± 1.0 ° C          $T_{modern}$ Katerloch = 8.8 ± 1.0 ° C

$\delta D_{modern}$ Obir = -79.9 ± 5.9 ‰          $\delta D_{modern}$ Katerloch = -69.1 ± 6.2 ‰

$\delta D/T_{gradient}$ Klagenfurt = 3.35 ± 1.40          $\delta D/T_{gradient}$ Graz = 2.66 ± 1.25

**Table A2: Paleotemperatures obtained from δD fluid inclusion data using the OBK-FIT transfer function. T modern refers to the temperature outside the cave. δD modern was corrected for elevation.**



**Data Availability**

The stable isotope data, the U/Th data and Fluid inclusions data that support the findings of this study will be made available as a download excel file in the supplement and later be integrated in the SISAL database.

**Author Contribution**

**C.H.** participated in the fieldwork, wrote the manuscript, performed most of the analytical work (stable isotopes, U/Th dating, fluid inclusions) and data analysis. G.K. assisted with the fluid inclusion analysis and provided manuscript feedback. C.S.
organised the fieldwork and contributed to the manuscript. H.Z. carried additional U/Th analysis and R.L.E and H.C. supported with U/Th analyses.

**Competing interests**

The authors declare that there are no conflicts of interest.

**Acknowledgements**

This study was funded by the Austrian Science Fund (FWF) grant P300040 to CS; and additional fund for the U/Th analysis by the NSFC grant 41888101 to HC. We thank Fritz Geisler and Harald and Andreas Langer for supporting our scientific work in Katerloch and Obir Caves. We would like to thank Tanguy Racine for his help during fieldwork and Kathleen Wendt for running preliminary ages on stalagmite OBI117, Marlene Steck for her help in the fluid inclusion stable isotope laboratory, Manuela Wimmer for her support with the calcite stable isotope measurements.

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
