# Peer review of "A paleoprecipitation and paleotemperature reconstruction of the Last Interglacial in the southeastern Alps"

_Climate of the Past, 2022_

## Author Response (AR1)

Dear editor,

We are grateful for the constructive comments and helpful suggestions by the reviewers. Below is a point-by-point response to the major comments and questions raised by each referee.

**Referee 1** - **Attila Demény**

**Abstract and onwards: δD is an old term, the new IUPAC definition requires the use of δ$^2$H instead. I suggest to correct it in the entire manuscript.**

Following the reviewer's suggestion, we changed the δD notation to δ$^2$H.

**Chapter titles: The expressions of „calcite stable isotope" and „fluid inclusion isotope" are a bit sloppy, the correct terms would be „stable isotope compositions of ...". The short titles „oxygen isotopes" and „carbon isotopes" should also be corrected for more correct terms, like „Stable oxygen isotope compositions". The reason is that although there is no radioactive oxygen isotope, this is not valid for carbon that has 14C.**

The chapter titles were updated to comply with the this comment.

**Fig. 2: black arrows would be more visible.**

The color was changed.

**Chapter 5.1.1., first sentence. The temperature of the cave would control the oxygen isotope composition of the speleothem calcite, but would not affect the δ18O value of the drip water. I think these two are mixed in the sentence.**

This sentence was corrected.

**Chapter 5.1.1., last sentence. This is called Replication Test, and the paper of Dorale and Liu (2009 should be cited. (Dorale J.A. – Liu Z. 2009: Limitations of Hendy test criteria in judging the paleoclimatic suitability of speleothems and the need for replication. – J. Caves Karst Stud., 71, pp. 73–80.)**

The sentence was updated with the citation.

**Chapter 5.1.2. In order to determine if biogenic activity or kinetic fractionation is the main factor in the δ13C values, these two should be compared by numbers. We saw that along lamina δ13C shifts can reach 4 ‰, which is not negligible. On the other hand, we see no information on the δ13C difference between winter and summer laminae.**

This is an important point. 4‰ is the highest value (not the average) we measured and is indeed not a negligible increase along a lamina. To account for this, we did not discuss (in terms of paleoclimate) changes in δ$^{13}$C values lower than 4‰.

We decided not to discuss the δ$^{13}$C difference between winter and summer laminae (of modern samples) because this difference might be related to anthropogenic modifications of the cave (and its air flow pattern) as a result of the show cave development during the mid-20$^{th}$ century, and hence not representative of the LIG. Additionally, the sampling resolution of the Katerloch LIG samples does not allow to discuss the isotopic signal on a seasonal scale.

**Chapter 5.2.1., last paragraph. I think the paper of Demény et al. (2021) could be cited here.**

Done.

**Line 340. There is a sudden change from the δ18O/T gradients to the δ2H/T values. I think the δ2H/T values are calculated from the δ18O/T ones using the LMWL equation, but this should be clarified in a bit more detailed discussion.**

This was clarified in the text.

**Line 357. Actually the MIS6-MIS5 glacial-interglacial δ2H change can be observed in the ABA-1 and the BAR-II records together (see Fig. 6).**

Thank you for pointing this out, we changed the text accordingly. We do say in the manuscript that there is a rise in $\delta^2H$ values of up to 25‰ across this glacial-interglacial transition seen in samples from caves on Melchsee-Frutt and from the combined record of Abaliget and Baradla caves, Hungary.

**Referee 2 – D. Genty**

**Section 2.1: Depths of cave galleries would be appreciated here.**

This information was added in the text: Banane system, Sandgang (OBI 117) is approx. 65 m below the surface; Banane system, entrance part (OBI 118) is approx. 20 m below the surface. Indische Grotte in the Obir show cave is approx. 45m below the surface, and Phantasiehalle in Katerloch is approx. 165 m below the surface.

**Section 2.3: Graphs with seasonality of rainfall quantity, d18O are needed. Also a map of air mass directions along the year to see the influence of the Mediterranean sea.**

We agree with the reviewer that a graph with seasonality of rainfall quantity and $\delta^{18}O$ would be a useful addition. This graph has been added in the supplement as Figure A3. However, we think that a map of air mass trajectories would be too complex to produce and will not add important information to the paper.

**Section 2.5: Indicate the duration of the dripping d18O monitoring for each station. Add chronicles. In figure 3 and 4, please put the same direction for the time scales.**

The duration of monitoring in Obir was almost 5 years with a visit every two months from summer of 1998, except for the period between March 2000 and December 2002 when the visiting frequency was increased to monthly (see more details in Spötl et al. 2005). In Katerloch, the monitoring interval was 2 months for a period of 2 years. The time scale of Figure 3 was updated accordingly. We are unsure what the reviewer means by "chronicles".

**Section 4.3.1: I agree that in figure 4, K4 and OBI117 do record the transition/onset (or part of), but note that OBI14 (green) and OBI99 (dark blue) cross cover the K4 sample but do not show large variations in d18O. Is it due to age uncertainties or to the way of how stalagmites from K cave record the transition in a different way ? Same question for the d13C, OBI14 and OBI99 do not show such**

**large changes, why ? From this figure it is clear that Obir samples isotope values are much more dampened than Katerloch ones.**

OBI14 and OBI99 do not show this prominent jump in $\delta^{18}O$ at Termination II, because they only started growing after this transition; the short "cross-cover" is simply due to the age uncertainties of the samples.

We agree that the $\delta^{18}O$ records of the Obir stalagmites seem more dampened than the Katerloch ones, possibly due to a longer residence time in the karst. The higher variability of the LIG $\delta^{13}C$ records from Katerloch compared to Obir is consistent with the seasonal variability of modern and Holocene stalagmites from this cave, which is controlled by seasonal changes in cave air pCO2 (as published by Boch et al., 2011). LIG speleothems from Obir lack such high-frequency variability, because prior to the artificial opening of these caves the exchange with the outside atmosphere was strongly reduced.

**Section 4.4: A graph with all the FI data from table 2 (and LIG samples) would be useful.**

Data from LIG are shown in Figure 4 already and we do not feel that a second graph is needed.

**Section 5.2.1: There is a kind of contradiction when you say that «only the stalagmite δD values were used because the δ18 OFI values in speleothems are influenced by non-climatic parameters (e.g., kinetic isotope fractionation)» while, line 283, you say that samples are close to the O isotopic equilibrium (Hendy test, replicability). Also, please consider the work of Dassié et al., 2018, on the different tests made on present day calcites and links with the parent drip water isotopes. I am not sure that there are O exchanges with matrix, and using dD brings additional uncertainties.**

Hendy tests in conjunction with the replication test (multiple overlapping stalagmites) support calcite precipitation close to O isotopic equilibrium. We cannot exclude an exchange of O isotopes between the water from the fluid inclusion and the host calcite, although this effect is possibly small, for example if the speleothem remained at temperatures close to the formation temperature, (Meckler et al., 2015; Uemura et al., 2020), but see also Demeny et al. (2016). This is not in contradiction to the use of $\delta^{2}H$ values of FI samples because we know that $\delta^{18}O$ values obtained at the Innsbruck FI setup are often inaccurate and we therefore do not use them. This was made more explicit in the text.

**Figure A5 – Middle : This is a nice view of a typical « competitive growth »**

We agree.

**Potential additional references:**

**For FI methods, replicability, mass/isotope effects etc.: Dassié E., Genty D., Noret A., Mangenot X., Massault M., Lebas N., Duhamel M., Bonifacie M., Gasparrini M., Minster B., Michelot J.L., 2018, A newly designed analytical line to examine fluid inclusion isotopic compositions in a range of carbonate samples, *Geochemistry, Geophysics, Geosystems*, 19, 1107-1122. https://doi.org/10.1002/2017GC007289**

**For rainfall and drip water isotope relationships: Genty, D., Labuhn, I., Hoffmann, G., Danis, P., Mestre, O., Bourges, F., Wainer, K., Massault, M., Van Exter, S., Regnier, E., Orengo, P., Falourd, S.,**

**Minster, B., 2014. Rainfall and cave water isotopic relationships in two South-France sites.** *Geochimica et Cosmochimica Acta* **131, 323-343.**

**For FI high-resolution records from the Holocene: Labuhn I., Genty D., Vonhof H., Bourdin C., Blamart D., Douville E., Ruan J., Cheng H., Edwards R.L., Pons-Branchu E., Pierre M. , 2015, A high-resolution fluid inclusion δ$^{18}$O record from a stalagmite in SW France: modern calibration and comparison with multiple proxies,** *Quaternary Science Review***, 110, 152-165.**

The second reference was added to the text.